# Viral Vectors in Gene Therapy: Where Do We Stand in 2023?

**DOI:** 10.3390/v15030698

**Published:** 2023-03-07

**Authors:** Kenneth Lundstrom

**Affiliations:** PanTherapeutics, CH1095 Lutry, Switzerland; lundstromkenneth@gmail.com

**Keywords:** viral vector, gene therapy, cancer, chronic disease, vaccines, preclinical models, clinical trials, approved drugs

## Abstract

Viral vectors have been used for a broad spectrum of gene therapy for both acute and chronic diseases. In the context of cancer gene therapy, viral vectors expressing anti-tumor, toxic, suicide and immunostimulatory genes, such as cytokines and chemokines, have been applied. Oncolytic viruses, which specifically replicate in and kill tumor cells, have provided tumor eradication, and even cure of cancers in animal models. In a broader meaning, vaccine development against infectious diseases and various cancers has been considered as a type of gene therapy. Especially in the case of COVID-19 vaccines, adenovirus-based vaccines such as ChAdOx1 nCoV-19 and Ad26.COV2.S have demonstrated excellent safety and vaccine efficacy in clinical trials, leading to Emergency Use Authorization in many countries. Viral vectors have shown great promise in the treatment of chronic diseases such as severe combined immunodeficiency (SCID), muscular dystrophy, hemophilia, β-thalassemia, and sickle cell disease (SCD). Proof-of-concept has been established in preclinical studies in various animal models. Clinical gene therapy trials have confirmed good safety, tolerability, and therapeutic efficacy. Viral-based drugs have been approved for cancer, hematological, metabolic, neurological, and ophthalmological diseases as well as for vaccines. For example, the adenovirus-based drug Gendicine^®^ for non-small-cell lung cancer, the reovirus-based drug Reolysin^®^ for ovarian cancer, the oncolytic HSV T-VEC for melanoma, lentivirus-based treatment of ADA-SCID disease, and the rhabdovirus-based vaccine Ervebo against Ebola virus disease have been approved for human use.

## 1. Introduction

Gene therapy has been defined as the supplementation, correction, or modification of malfunctional genes by functional equivalents for therapeutic correction of the absence or reduced levels of gene expression activity [1]. A broader definition considers oligonucleotide- [2] and RNA interference (RNAi)-based gene silencing [3], immunotherapy, especially cancer immunotherapy, and vaccine development as gene therapy [4]. More recently, stem cell technologies [5], chimeric antigen receptor (CAR) T-cell therapy [6], and Clustered Regularly Interspaced Short Palindromic (CRISPR) [7], providing unprecedent possibilities for gene replacement and gene editing, have also received gene therapy status.

Viral vectors have played a central role in gene therapy because of their superior gene delivery capacity compared to non-viral vectors. Moreover, the virus-based transgene expression, depending on the needs, for both short-term and long-term duration can be achieved. For example, for cancer gene therapy, short-term high-level transgene expression is advantageous, whereas for chronic diseases such as hemophilia, long-term transgene expression is necessary. However, the application of viral vectors requires a higher biosafety level compared to non-viral vectors due to the risk of spread of virus progeny, especially in the case of using replication-competent and oncolytic viruses. Other factors of importance are the regulation and termination of transgene expression. The history of gene therapy has been impacted by some tragic events, which was a setback for its proclaimed status as “the medicine of the future”. Although the retrovirus-based treatment of children with X-linked severe combined immunodeficiency (SCID-X1) was successful, the insertion of the therapeutic gene into the LMO2 proto-oncogene region of the genome led to the development of leukemia in a few patients [8]. In another case, inadequate planning and execution of clinical protocols for adenovirus-based treatment of the non-life-threatening ornithine transcarbamylase (OTC) deficiency resulted in the death of an 18-year-old patient [9]. These two incidents in the 1990s had a dramatic impact on the field of gene therapy, which put it more or less on hold until a renaissance occurred in recent years. However, during the years, efforts to develop more efficient and safer viral vectors continued, which has facilitated the return of gene therapy to the front of innovative drug and vaccine development. Although enormous progress has also been made in the area of non-viral vectors and their applications in gene therapy, the focus in this review is uniquely on viral vector systems and their utilization in preclinical studies and clinical trials.

## 2. Viral Vector-Based Delivery

Different types of viral vectors based on both DNA and RNA viruses have been engineered for gene therapy applications (Figure 1). The choice of vector is, to a large extent, affected by the disease indication, and need of short-term expression for acute diseases such as infectious diseases and cancers, and long-term expression required for chronic diseases. In the former case, high-level transient expression from replication-deficient viral vectors can provide therapeutic efficacy [10]. In the latter case, long-term expression is often achieved by extrachromosomal presence or chromosomal integration of the viral vector/transgene for extended therapeutic activity. Typically, replication-deficient and non-integrating vector systems are only capable of providing long-term transgene expression in post-mitotic tissues. In any viral vector-based gene therapy application, safety is of utmost importance [11]. Obviously, long-term treatment and presence of viral vector and/or transgene sequences in the host genome demands special requirements related to integration site, control of expression levels, and pharmacokinetics of the therapeutic product. In the context of cancer gene therapy, oncolytic viruses, which specifically replicate in tumor cells leading to their killing, have been evaluated as such, or as delivery vectors for anti-tumor genes both in vitro and in vivo [12]. A comprehensive description of various types of viral vectors is presented below and summarized in Table 1.

### 2.1. Adenovirus Vectors

Since the advent of gene transfer in mammalian cells, adenoviruses (Ad) vectors have been commonly used as viral delivery vehicles [13]. They are non-enveloped viruses possessing a double-stranded DNA (dsDNA) genome with a packaging capacity of up to 7.5 kb foreign DNA. However, Ad shuttle vectors have been engineered for accommodation of up to 14 kb of foreign DNA [14]. The first-generation Ad vectors were hampered by strong immunogenicity despite removal of the E1/E3 genes from the genome [15]. However, the immunogenicity has been reduced significantly in replication-deficient second- and third generation Ad vectors [16]. High-capacity third-generation adenovirus (HC-Adv) vectors, also known as helper-dependent gutless vectors, have the capacity to accommodate up to 37 kb of foreign DNA [17]. Moreover, replication-competent oncolytic adenoviruses have been developed for specific replication in tumor cells, resulting in the killing of tumor cells [18]. The engineering of packaging cell lines has facilitated large-scale GMP-grade Ad vector production [19]. Ad vectors provide persistent extrachromosomal transgene expression lasting for at least one year despite no integration into the host genome [20]. Moreover, a follow-up study in non-human primates showed transgene expression, although reduced to 10% of peak values, up to 7 years without any long-term adverse effects [21].

### 2.2. Adeno-Associated Virus Vectors

The small non-enveloped single-stranded DNA (ssDNA) adeno-associated virus (AAV) can only accommodate 4 kb of foreign DNA [22], although, the packaging capacity has been improved by constructing fragmented, overlapping, or trans-splicing Dual AAV vectors [23,24]. AAV vectors generally do not cause toxic or pathogenic responses. However, repeated administration of AAV vectors has generated strong immune responses, reducing the efficacy of delivery and transgene expression [25]. This problem has been addressed by applying different AAV serotypes for each AAV re-administration. An alternative approach has been to utilize exosome-associated AAV (Exo-AAV), which has supported the application of reduced AAV doses resulting in reduced immune responses against the AAV capsid protein [26]. Moreover, Exo-AAV8 vectors have demonstrated long-term liver-directed gene transfer [27]. AAV vectors can transduce both dividing and non-dividing cells and usually remain in an extrachromosomal state, although integration of AAV-delivered genes into the host genome has been reported [28]. In fact, 30-fold higher AAV integration frequency was obtained by the introduction of 28S ribosomal DNA homology sequences in AAV vectors, which might contribute to superior treatment of genetic diseases [29].

### 2.3. Herpes Simplex Virus Vectors

The large herpes simplex viruses (HSV) are enveloped dsDNA viruses, which cause latent infection in neural ganglia [30]. The engineering of HSV expression vectors has resulted in long-lasting transgene expression [31]. The linear HSV forms a circularized viral episome in the nucleus and remains extrachromosomal without integration [32]. HSV vectors have an excellent capacity of accommodating more than 30 kb of foreign DNA [33]. Engineered HSV amplicons are able to package 150 kb of foreign genetic material [34]. However, HSV vectors have been associated with relatively strong cytopathogenicity, which has been addressed by the deletion of non-essential genes in the HSV genome [35]. Furthermore, the introduction of micro-RNA sequences (miR-145) in the HSV ICP27 gene has generated oncolytic HSV vectors, which can selectively reduce cell proliferation in non-small cell lung cancer (NSCLCs) cells [36]. Efficient HSV packaging systems have been engineered, such as the helper virus-free system for the HSV amplicon using an ICP27-deleted, oversized HSV-1 DNA in a bacterial artificial chromosome (BAC) [37].

### 2.4. Retrovirus and Lentivirus Vectors

The enveloped single-stranded RNA (ssRNA) retroviruses (RVs) possess a packaging capacity of 8 kb of foreign sequences [38]. The special feature of RVs comprises their reverse transcriptase activity, which allows the production of dsDNA copies of the RNA genome for integration into the host genome [39]. The chromosomal integration is advantageous for long-term transgene expression, although random integration has been of concern, even resulting in leukemia development in treated SCID-X1 patients [8]. For this reason, self-inactivating γRV (SIN-γRV) vectors have been engineered, which have proven safe with no cases of adverse integration or leukemia observed in clinical trials [40]. However, adenosine deaminase deficient severe combined immunodeficiency (ADA-SCID) seems to differ from other inherited immunodeficiencies, as insertional oncogenesis is rare after γRV treatment [41]. For example, none of the 10 patients in a clinical study developed leukemia [42], and among a total of 50 ADA-SCID patients treated with γRV, only one showed clinical evidence of leukemia [43]. Packaging cell lines have also been engineered for RV vectors to support large-scale production of GMP-grade materials [44]. One serious limitation of gene therapy applications for RV vectors is their capability to only transduce dividing cells and not non-dividing cells.

In contrast, lentivirus (LV) vectors, which also belong to the family of retroviruses, can transduce both dividing and non-dividing mammalian cells [45]. Otherwise, LV vectors share the same features with RVs of an ssRNA genome and a capacity of carrying 8 kb of foreign genetic material. Importantly, LV vectors show low cell cytotoxicity and due to their ”semi-random” chromosomal integration provide improved biosafety for clinical applications, although some adverse events and insertional oncogenesis have been reported [46]. For example, modification of integration of human immunodeficiency virus-1 (HIV-1) by the fusion of the C-terminal HIV integrase-binding region of the LEDGF/p75 protein to the N-terminal chromodomain of heterochromatin protein-1 alpha (HP1 alpha) reduced the number of integration events [47]. Expression systems for non-human LV vectors such as simian immunodeficiency virus (SIV) [48], feline immunodeficiency virus (FIV) [49], and equine infectious anemia virus (EIAV) [50] have been engineered. LV producer cell lines have been designed to support large-scale production [51]. However, the low titers obtained, and residual toxicity have compromised their utilization [51].

### 2.5. Alphavirus Vectors

Alphaviruses are enveloped viruses with an ssRNA genome of positive polarity and a packaging capacity of 8 kb of foreign genetic material [52,53]. The positive polarity of alphaviruses allows the direct translation of viral RNA in the host cell cytoplasm. Alphaviruses possess a special feature of RNA self-replication, which generates extreme levels of transgene expression. The nature of expression is transient due to the rapid degradation of the alphavirus ssRNA. Alphavirus vectors can be used as recombinant particles, naked or liposome encapsulated RNA replicons, or plasmid DNA-based replicons [54]. Expression systems have been developed for Semliki Forest virus (SFV) [55], Sindbis virus (SIN) [56], and Venezuelan equine encephalitis virus (VEE) [57]. Moreover, naturally occurring oncolytic M1 viruses [58] and engineered oncolytic SFV vectors [59] have been used for cancer therapy.

### 2.6. Flavivirus Vectors

Similar to alphaviruses, flaviviruses are enveloped ssRNA viruses of positive polarity and therefore possess the feature of self-replicating RNA, providing high levels of transient transgene expression and the flexibility of using recombinant viral particles, RNA replicons and DNA replicons [60]. The packaging capacity of flaviviruses is approximately 6 kb. Kunjin virus (KUN) [60], West Nile virus (WNV) [61], Dengue virus (DENV) [62], tick-borne encephalitis virus (TBEV) [63], yellow fever virus (YFV) [64], and Zika virus (ZIKV) [65] have been subjected to the engineering of expression systems. In support of large-scale KUN [66] and TEBV [63] vector production, packaging cell lines have been engineered.

### 2.7. Measles Virus Vectors

The enveloped measles viruses (MVs) carry an ssRNA genome of negative polarity [67]. For this reason, the MV RNA first needs to be copied as a positive strand RNA template for self-replication of RNA in the host cytoplasm before being translated [68]. Approximately 6 kb of foreign genetic material can be introduced into MV vectors. Technologies for reverse genetics [69] and packaging cell lines [70] have been established. Oncolytic MV strains such as MV Hu-191 [71] and MV Schwartz [72] have also been used for cancer therapy.

**Table 1 viruses-15-00698-t001:** Examples of viral vectors used for gene therapy applications.

Virus	Genome	Insert Size	Advantages and Limitations
**Adenovirus**			
Ad5	dsDNA	<7.5 kb	Broad host range (dividing and non-dividing cells) [13]
Ad26			Excellent packaging capacity of HC-Adv [17]
ChAd			Persistent expression, no chromosomal integration [20]
HC-AdV		37 kb	Strong immunogenicity [14], reduced for gutless Ad [16]
			Oncolytic Ad vectors for tumor targeting and killing [18]
			Pre-existing immunity in humans [13]
			Packaging cell lines for large-scale GMP production [19]
**AAV**			
AAV2, 3	ssDNA	4 kb	Relatively broad host range [22]
AAV5, 6			Limited packaging capacity [22] improved by Dual AAV vectors [23,24]
AAV8, 9			Strong immune response after AAV re-administration, which could be reduced by re-administration with different AAV serotypes [25]
Dual AAV			Exo-AAV vectors have reduced immunogenicity, providing liver-targeted transgene expression [26,27]
Exo-AAV			Generally, AAV remains in an extrachromosomal state [28]
**HSV**			
HSV-1	dsDNA	>30 kb	Broad host cell range [31], excellent [33], extreme for HSV amplicons [34] foreign DNA packaging capacity
HSV-2			Long-lasting transgene expression from extrachromosomal circular HSV DNA [32]
HSV amplicons		150 kb	Deletion of non-essential HSV genome reduces cytotoxicity [35]
			Engineering of oncolytic HSV by introduction of miR145 [36]
			Engineering of helper virus-free packaging system [37]
**γ** **-Retrovirus**			
MMSV	ssRNA	8 kb	Restricted host range, only dividing cells [38]
MSCV			Good packaging capacity of foreign genetic material [38]
SIN-γRV			Chromosomal integration due to reverse transcriptase activity [39]
			Random integration causing leukemia [8]
			Targeted integration with self-inactivating vector [40]
			Packaging cell lines for large-scale production [44]
**Lentivirus**			
HIV-1	ssRNA	8kb	Broad host range, including non-dividing cells [45]
HIV-2			Good capacity to accommodate foreign genetic material [45]
SIV			Non-random chromosomal integration [46]
FIV			Non-human LV vectors available [47,48,49,50]
EIAV			Producer cell lines engineered for LV vectors [51]
**Alphavirus**			
SFV, SIN,	ssRNA	8 kb	Extremely broad host range, risk of neurovirulence [52]
VEE, M1			Good packaging capacity [53]
			RNA self-replication leading to extreme transgene expression [52]
			Low immunogenicity of alphaviruses [52]
			Transient expression not applicable for chronic diseases, but good for acute diseases and vaccines [52]
			Flexibility to use viral particles, RNA and DNA replicons for delivery [54]
			Oncolytic alphaviruses for cancer therapy [58,59]
**Flavivirus**			
KUN, WNV,	ssRNA	6 kb	Broad host range, relatively good packaging capacity [60]
DENV, TBEV			RNA self-replication leading to high transgene expression [60]
YFV, ZIKV			Flexibility to use viral particles, RNA and DNA replicons for delivery [60]
			Efficient packaging cell lines for KUN [66] and TBEV [63]
**Measles virus**			
MV	ssRNA	6 kb	Broad host range, relatively good packaging capacity [67]
			Positive strand RNA template needed for translation [68]
			Development of reverse genetics [69] and packaging cell lines [70]
			Oncolytic MV strains for cancer therapy [71,72]
**Rhabdovirus**			
VSV	ssRNA	6 kb	Broad host range, relatively good packaging capacity [73]
RABV			Positive strand RNA template needed for translation [73]
Maraba			Reverse genetics systems [74]
			Oncolytic rhabdoviruses for cancer therapy [75,76]
			Vaccinia-free packaging cell lines [77]
**NDV**			
NDV	ssRNA	4 kb	Broad host range, modest packaging capacity [78]
			Reverse genetics systems available [79]
			Oncolytic NDV for killing of tumor cells [79]
**Poxvirus**	dsDNA	>30 kb	
VV			Broad host range [80]
Avipox			Excellent packaging capacity [80]
			Tumor-selective replication-competent VV [81]
**Picornavirus**			
CVA21	ssRNA	6 kb	Relatively broad host range [82]
CVB3			Relatively good packacking capacity despite the small size [82]
PV-1			No chromosomal integration [82]
			Applications for gene therapy and vaccines [83,84]
**Reovirus**			
Reovirus-3	dsRNA	ND	Oncolytic activity in different types of cancer cells [85]
			Reoviruses replicate preferentially in Ras activated tumor cells [86]
			Combination therapy with radio-, chemo-, and immunotherapy [87]
			Endoplasmic reticular stress-mediated apoptosis in cancer cells [88]
**Polyoma virus**			
SV40	dsDNA	17.7 kb	Superb packaging capacity of 17.7 kb for SV40 with small genome [89]
			Vero cell-based SV40 packaging system [90]
			Inhibition of tumor cell progression [91]

AAV, adeno-associated virus; Ad, adenovirus; CVA21, coxsackievirus A21; CVB3, coxsackievirus B3; DENV, Dengue virus; dsDNA, double-stranded DNA; dsRNA, double-stranded RNA; Exo-AAV, exosome-associated AAV; FIV, feline immunodeficiency virus; HC-Adv, high-capacity Ad gutless vector; HIV, human immunodeficiency virus; HSV, herpes simplex virus; KUN, Kunjin virus; M1, oncolytic alphavirus; MMSV, Moloney murine sarcoma virus; MSCV, murine stem cell virus; ND, not determined; NDV. Newcastle disease virus; PV-I, poliovirus-1; SFV, Semliki Forest virus; SIN, Sindbis virus; SINγRV, self-inactivating gamma retrovirus; SIV, simian immunodeficiency virus; ssDNA, single-stranded DNA; ssRNA, single-stranded RNA; TBEV, tick-borne encephalitis virus; VEE, Venezuelan equine encephalitis virus; VV, vaccinia virus; WNV, West Nile virus; YFV, yellow fever virus; ZIKV, Zika virus.

### 2.8. Rhabdovirus Vectors

Also, rhabdoviruses are enveloped ssRNA viruses with a negative-stranded genome [73]. Reverse genetics methods have been applied for the generation of rhabdovirus expression systems [74]. Generally, 6 kb of foreign sequences can be accommodated in rhabdovirus vectors [73]. Expression systems have been engineered for rabies virus (RABV) [92], vesicular stomatitis virus (VSV) [93], and Maraba virus [94]. The majority of the oncolytic rhabdovirus vectors are based on VSV [75] and Maraba virus [76]. Moreover, vaccinia-free packaging cell lines have been established for VSV [77].

### 2.9. Newcastle Disease Virus Vectors

The enveloped negative-stranded ssRNA Newcastle disease virus (NDV) has a limited packaging capacity of only 4 kb of foreign genetic material [78]. However, this has not been a major issue as NDV vectors possess oncolytic activity and specifically replicate in tumor cells, resulting in efficient cell killing and tumor eradication [95]. Oncolytic NDV vectors have been used for cancer therapy in both preclinical animal models and clinical trials [76]. Reverse genetics have also been used for the NDV-73 T strain to modify the cleavage site of the fusion (F) protein, which decreased the pathogenicity in chicken without reducing the potency of tumor cell killing [96].

### 2.10. Poxvirus Vectors

Poxviruses are large, enveloped dsDNA viruses [80], which show an outstanding packaging capacity of more than 30 kb of foreign DNA. Among poxviruses, vaccinia virus (VV) vectors have been frequently used for prophylactic and therapeutic applications in the fields of infectious diseases and cancers [97]. Engineering of tumor-selective replication-proficient VV vectors has proven an attractive approach for cancer therapy [81]. In the context of avian poxviruses such as the non-replicating avipox virus, good biosafety standards have been achieved for non-avian species [98].

### 2.11. Picornavirus Vectors

The small non-enveloped picornaviruses contain an ssRNA genome and are capable of introducing up to 6 kb of foreign genetic material despite their small size [81]. Both coxsackievirus A21 (CVA21) [82] and the attenuated coxsackievirus B3 (CVB3) [83] have proven useful for gene therapy and vaccine development. Expression systems have also been engineered for the PV-1 poliovirus [84].

### 2.12. Reovirus Vectors

The enveloped dsRNA reoviruses possess oncolytic activity, showing killing of different types of cancer cells [85]. It has been documented that reoviruses replicate preferentially in tumor cells with activated genes of the Ras family or Ras-signaling pathway, which can be found in 60–80% of human malignancies [86]. Reovirus vectors have been demonstrated to invoke immune stimulation for reversing tumor-induced immunosuppression and promotion of anti-tumor immune responses [99]. Reoviruses have also been combined with radiotherapy, chemotherapy, immunotherapy, and surgery for cancer treatment [87]. Moreover, reovirus serotype 3 (Reolysin^®^) induces endoplasmic reticular stress-mediated apoptosis in in vivo models of pancreatic cancer [88].

### 2.13. Polyoma Virus Vectors

Although, the small non-enveloped dsDNA viruses possess a genome of only 5 kb, for example, the simian virus 40 (SV40) can package 17.7 kb of foreign DNA [89]. Packaging of virus-like particles (VLPs) containing no SV40 wildtype sequences can be carried out in vitro. Additionally, Vero cell-based packaging systems have been engineered for SV40 [90]. SV40 vectors have demonstrated successful delivery of anti-viral agents, DNA vaccines, suicide, chemoprotective, and anti-angiogenic genes for successful inhibition of tumor cell progression [91].

## 3. Gene Therapy Applications

Due to the many gene therapy studies conducted with viral vectors, it is only possible to provide an overview here through examples from preclinical studies and clinical trials for various diseases. The examples are selected to cover most disease indications using different types of viral vectors without indicating any preference of vector choice. The findings are also summarized in Table 2, Table 3, Table 4 and Table 5.

### 3.1. Cancer

Different types of cancers have been frequently targeted by viral vector-based gene therapy and immunotherapy, and the potentially straightforward tumor killing with no need for long-term transgene expression. In addition to intratumoral administration, tumor targeting by specifically designed vectors [100], utilization of tumor-specific promoters [101], and application of oncolytic viruses [9] have been tested (Table 2).

**Table 2 viruses-15-00698-t002:** Preclinical and clinical examples of viral vectors applied for cancer therapy.

Viral Vector	Phase	Findings
**Breast cancer**		
HSV-HF10	Pre	Substantial tumor regression, prolonged survival in mice [102]
Reolysin + anti-PD1	Pre	Superior tumor regression, prolonged survival in mice after combination [103]
M1	Pre	Targeting and killing of 4T1 mammary tumors in mice [104]
PANVAC	Phase I	SD in 4 patients, complete response in 1 patient [105]
**Gliomas**		
M1	Pre	Specific targeting of C6 glioma cells [106]
M1	Pre	Replication of M1 in gliomas in mice, rats, and macaques [107]
SFV-IL-12	Pre	87% reduction of RG2 glioma size in rats [108]
SFV-VA	Pre	100% eradication of small, 50% eradication of large tumors in mice [59]
RRV Toca 511-CD	Pre	Prolonged survival in mice with orthotopic gliomas [109]
m-ZIKV	Pre	Prolonged survival in mice with implanted glioblastomas [65]
MV-CEA	Phase I	Trial in progress in patients with recurrent glioblastoma [110]
RRV Toca 511	Phase I	Prolonged survival of 13.6 months in HGG patients [111]
RRV Toca 511	Phase II/III	No improvement in overall survival in HGG patients [112]
**Colon cancer**		
KUN-GM-CSF	Pre	Cure of >50% of mice with CT26 colon tumors [113]
VSV(M51R)	Pre	Reduced luciferase expression in tumors, prolonged survival in mice [114]
M-LPO	Pre	Superior oncolytic activity in mice [115]
SFV-LacZ RNA	Pre	Protection against tumor challenges in mice after a single injection of RNA [116]
VEE-CEA	Phase I	Antigen-specific responses and prolonged survival in colorectal cancer patients [108,117]
vvDD	Phase I	Th1-biased immune responses against vvDD and tumors in patients [118]
**Melanoma**		
KUN-GM-CSF	Pre	Significant tumor regression, 67% of mice tumor-free [113]
NDV-IL12/IL15	Pre	Superior survival after NDV-IL15 compared to NDV-IL12 in mice [119]
VSV-LCMV-GP	Pre	Significant tumor regression, prolonged survival in melanoma models [120]
VSV-XN2-ΔG	Pre	Strong tumor regression in mice [121]
CVA21-ICAM-DAF	Pre	Tumor regression, reduced tumor burden in mouse melanoma model [122]
MG1-hDCT + Ad	Pre	Ad-hDCT prime-Maraba MG1-hDCT booster elicited immune responses [94]
HSV-HF10 + CTLA4	Phase III	Good safety and antitumor activity in patients [102]
HSV T-VEC	Phase II/III	Good tolerance, promising therapeutic effect in melanoma patients [123]
HSV T-VEC	Approval	Approved for treatment of advanced melanoma in the US, Europe, Australia [124]
**Pancreatic cancer**		
Adsur-SYE	Pre	Complete tumor regression in mice [125]
PANVAC	Pre	Superior immune response in pancreatic mouse cancer models [126]
SV40-hRT-SST2	Pre	Long-term inhibition of tumors in Capan-1 mouse tumor model [91]
HSV-HF10	Phase I	PR in 3 patients, SD in 4 patients, PD in 9 patients [127]
**Ovarian**		
VSV-LCMP-GP	Pre	Tumor regression in ovarian cancer mouse models [128]
VSV-LCMP-GP + Rux	Pre	Superior therapeutic activity after combination therapy [128]
VSVMP-p DNA	Pre	87–98% tumor regression in ovarian mouse cancer models [129]
SIN AR339	Pre	Ovarian cancer cell killing, tumor regression in mice [130]
MV-CEA	Phase I	SD in all 9 patients, overall survival twice to the expected time [131]
**Prostate**		
MV-CEA	Pre	Delay of tumor growth, prolonged survival [132]
MV-sc-Fv-PSMA	Pre	Specific killing of prostate tumors, enhanced by radiation [133]
MV + MuV	Pre	Superior antitumor activity, survival after combination therapy [134]
VEE-PSMA	Pre	Strong Th1-biased immune responses in mice [135]
VEE-mSTEAP	Pre	Prime immunization with DNA, booster with VEE specific immunogenicity [136]
VEE-PSCA	Pre	Long-term survival in 90% of TRAMP mice [137]
VV-GLV-1h123-NIS	Pre	Inhibition of tumor growth, prolonged survival in prostate cancer models [138]
VSV-PSMA	Phase I	Good safety, disappointingly weak immune responses [139]

Adsur-SYE, adenovirus vector with survivin promoter, pancreatic cell-targeting ligand SYENFSA; CD, yeast cytosine deaminase; CTLA4, anti-CTLA-4 antibody; DAF, decay accelerating factor; HGG, high grade glioma; HSV, herpes simplex virus; ICAM-1, intercellular adhesion molecule-1; KUN. Kunjin virus; LCMP-GP, lymphocytic choriomeningitis virus-glycoprotein; M1, oncolytic alphavirus; M-LPO, liposome-encapsulated M1 alphavirus; mSTEAP, mouse six-transmembrane epithelial antigen of the prostate; m-ZIKV, mouse-adapted Zika virus; MV, measles virus; NDV, Newcastle disease virus; PD, progressive disease; PR, partial response; Pre, preclinical studies; PSCA, prostate stem cell antigen; PSMA, prostate-specific membrane antigen; RRV replicating retrovirus; Rux, ruxolitinib; SD, stable disease; SFV, Semliki Forest virus; SIN, Sindbis virus; TRAMP, transgenic adenocarcinoma of the prostate; VSV, vesicular stomatitis virus; vvDD, oncolytic vaccinia virus.

For example, substantial tumor regression and prolonged survival were observed in mouse breast tumor models after treatment with an HSV-HF10 vector [102], and the co-treatment with a reovirus vector and checkpoint inhibitor PD-1 antibody [103]. The oncolytic M1 alphavirus efficiently targeted and killed 4T1 mammary tumors in mice [104]. In one example of a clinical study, the PANVAC vaccine based on VV and fowl pox was subjected to a phase I trial in heavily pre-treated breast cancer patients [105]. Stable disease (SD) was observed in four patients and one patient showed a complete response [105]. In the context of gliomas, the M1 alphavirus showed specific targeting of C6 glioma cells [106] and replication in gliomas in mice, rats, and macaques [107]. Moreover, SFV particles expressing interleukin-12 (SFV-IL-12) were administered via an implanted canula, which reduced RG2 gliomas by 87% in rats [108]. In another study, the replication competent SFV VA7 showed strong killing of human glioma cells, and intravenous administration in BALB/c mice completely eradicated 100% of small and 50% of large subcutaneous U87 tumors [59]. In the case of RV vectors, the replicating retroviral vector (RRV) Toca 511 carrying the yeast cytosine deaminase (CD) provided extended survival in mice implanted with orthotopic gliomas [109]. ZIKV has demonstrated specific targeting and killing of glioblastoma stem cells (GSCs), and administration of the mouse-adapted ZIKV (m-ZIKV) strain prolonged survival substantially in mice with implanted glioblastomas [65]. Moreover, MV particles expressing the carcinoembryonic antigen (CEA) [140] have been subjected to a phase I trial in patients with recurrent glioblastoma multiforme [110]. In a phase I trial, patients with recurrent or progressive high-grade glioma (HGG) who received the RRV Toca 511 vector showed a statistically relevant extended survival of 13.6 months [111]. In contrast, the overall survival was not prolonged in phase II/III trials in HGG patients [112].

KUN-based expression of the granulocyte macrophage-colony stimulating factor (GM-CSF) resulted in cure in more than 50% of CT26 colon tumor-bearing mice [113]. In another approach, the oncolytic VSV(M51R) strain was administered intraperitoneally into BALB/c mice carrying luciferase-expressing CT26 tumors, which resulted in eradication of tumors demonstrated by reduced luciferase expression and prolonged survival of mice [114]. M1 alphavirus particles encapsulated in liposomes (M-LPO) were able to inhibit the growth of colorectal LoVo and liver Hep3B cancer cells [115]. Moreover, intravenous administration of M-LPO reduced the production of M1-specific neutralizing antibodies in mice, resulting in superior oncolytic activity [106]. In an interesting approach, only a single injection of 0.1 μg of naked SFV-LacZ replicon RNA provided protection in mice with implanted CT26 colon tumors against tumor challenges [116]. Additionally, therapeutic activity and prolonged survival were found in mice with pre-existing tumors [116]. In the case of clinical trials, VEE-CEA particles were administered to stage III and IV colorectal cancer patients in a phase I trial [117]. It was found that antigen-specific immune responses were detected in both stage III and IV patients, and the overall survival was extended. In another phase I trial, patients with advanced colorectal cancer were subjected to oncolytic vvDD poxvirus particles, which elicited potent Th1-biased immune responses against vvDD and tumors [118].

Melanoma has been frequently targeted for gene therapy applications of viral vectors. For example, KUN-GM-CSF particles generated significant tumor regression and cured 67% of mice with B16-OVA melanomas [113]. In another approach NDV vectors were applied for the expression of IL-12 and IL-15 [119]. Intratumoral administration of NDV-IL12 and NDV-IL15 into a mouse melanoma model suppressed tumor growth. NDV-IL15 was superior, showing 26.6% higher survival rate compared to NDV-IL12 [119]. In another study, chimeric VSV particles expressing the lymphocytic choriomeningitis virus glycoprotein (LCMV-GP) showed significant tumor regression and prolonged survival in syngeneic melanoma tumor models [120]. In another study on VSV, strong tumor regression was seen in C57BL/6 mice implanted with B16-OVA melanomas after subcutaneous injection of an oncolytic VSV vector [121]. In the context of picornaviruses, a single subcutaneous injection of CVA21 particles expressing the intercellular adhesion molecule-1 (ICAM-1) and the decay-accelerating factor (DAF) resulted in tumor regression and reduced tumor burden in a mouse melanoma model [122]. The oncolytic Maraba MG1 strain expressing the human dopachrome tautomerase (hDCT) neither elicited antitumor immune responses nor therapeutic activity in mice with B16-F10 metastases [94]. However, prime immunization with Ad-hDCT followed by a booster immunization with Maraba MG1-hDCT elicited strong immune responses [94]. In contrast, the Maraba MG1 strain provided a long-lasting cure in sarcoma-bearing mice, and protection against challenges with sarcoma tumors [79]. In a phase III study, HSV-HF10 was combined with the checkpoint inhibitor anti-CTLA-4 antibody, demonstrating a good safety profile and antitumor activity in patients with non-resectable or metastatic melanoma [102]. HSV vectors, especially the oncolytic talimogene laherparevec (HSV T-VEC) vector expressing GM-CSF, have been assessed in Phase II and III clinical trials, showing a tolerable adverse event profile and promising therapeutic efficacy superior to GM-CSF therapy [123]. However, responses in visceral metastases have been modest. HSV T-VEC has been approved for the treatment of advanced melanoma in the US, Europe, and Australia [124].

Due to the aggressive nature and difficulty to treat pancreatic cancer, gene therapy efforts have been welcomed as an alternative strategy. For example, administration of Ad vectors containing the survivin promoter and the pancreatic cancer cell-targeting ligand SYENFSA (SYE) resulted in complete regression of pancreatic neuroendocrine tumors (PNETs) in mice [125]. Related to poxviruses, a heterogenous prime-boost strategy applying the PANVAC system for VV and fowl pox vectors elicited enhanced immune responses in pancreatic mouse cancer models [126]. A replication-competent SV40 vector carrying the tumor-specific human telomerase (hTR) RNA promoter and the somatostatin receptor tumor-suppressor 2 (SST2) gene showed long-term inhibition of tumor growth in the Capan-1 pancreatic mouse tumor model [91]. In a phase I trial, the oncolytic HSV-HF10 was administered intratumorally to patients with non-resectable locally advanced pancreatic cancer, showing partial response (PR) in three patients, SD in four patients, and progressive disease (PD) in nine patients [127].

In the case of ovarian cancer, the VSV-LCMV-GP showed tumor regression in subcutaneous and orthotopic ovarian cancer mouse models [128]. Moreover, the therapeutic efficacy was improved by co-administration of VSV-LCMV-GP and the JAK1/2 inhibitor ruxolitinib [128]. Application of the liposome-encapsulated VSVMP-p DNA vector expressing the VSV membrane (M) protein for intraperitoneal injection in mice reduced the tumor weight by 90%, and prolonged survival of mice with implanted ovarian tumors [141]. Moreover, the ovarian tumor growth was inhibited by 87–98% [129]. In another study, intraperitoneal administration of the oncolytic SIN AR339 vector resulted in ovarian cancer cell killing and tumor regression in mice [130]. In a clinical setting, MV-CEA particles were evaluated in a phase I trial in patients with recurrent ovarian cancer [131]. No dose-limiting toxicity was associated with the treatment, and SD was achieved in all nine treated patients. Moreover, the median overall survival was 12.15 months, which is twice the expected time.

Related to prostate cancer, intratumoral administration of MV-CEA particles delayed tumor growth and prolonged survival in PC-3 prostate tumor-bearing mice [132]. In another study, an MV vector expressing a single-chain antibody (sc-Fv) specific for the extracellular domain of the prostate-specific membrane antigen (PSMA) was administered to mice with LNCaP and PC3-PSMA prostate tumors [133]. MV-sc-Fv-PSMA provided specific infection and killing of PSMA-positive prostate cancer cells, which was further enhanced by radiation therapy. Co-administration of oncolytic MV and mumps virus (MuV) vectors showed superior antitumor activity, and prolonged survival in mice with PC-3 prostate tumors compared to administration of either MV or MuV vectors alone [134]. In another approach, VEE-based expression of the prostate-specific membrane antigen (PSMA) elicited strong PSMA-specific immune responses in BALB/c and C57BL/6 mice [135]. A single immunization induced strong T- and B-cell responses, which were Th1-biased. Moreover, a booster immunization with VEE particles expressing the mouse six-transmembrane epithelial antigen of the prostate (mSTEAP) 15 days after a prime immunization with gold-coated conventional pcDNA-3-mSTEAP plasmids elicited specific immune responses against mSTEAP, a modest but significant delay of tumor growth, and prolonged the overall survival of mice [136]. Moreover, administration of VEE particles expressing the prostate stem cell antigen (PSCA) resulted in long-term survival in 90% of transgenic adenocarcinoma of the prostate (TRAMP) mice [137]. In addition, administration of the VV GLV-1h123 vector expressing the sodium iodide symporter (NIS) gene provided significant inhibition of tumor growth, and extended survival time in prostate cancer mouse models [138]. In the context of clinical evaluation, a phase I trial was conducted in patients with castration resistant metastatic prostate cancer (CRPC) with VEE-PSMA particles [139]. Although the procedure showed good safety standards, the PSMA-specific immune responses were disappointingly weak.

### 3.2. Cardiovascular Diseases

Gene therapy-based applications for cardiovascular diseases have mainly focused on Ad and AAV vectors (Table 3). For example, expression of the sarcoplasmic reticulum Ca^2+^ ATPase (SERCa2a) by an Ad vector restored both systolic and diastolic heart functions to normal levels in a rat model of heart failure [142]. Ad-SERCa2a also managed to improve coronary blood flow, and reduced cardiomyocyte size in a rat model for type 2 diabetes [143]. SERCa2a has also been expressed from AAV-1 vectors leading to increased coronary blood flow in a pig model [144]. Moreover, LV-based expression of SERCa2a provided protection against left ventricular dilation, improved systolic and diastolic functions, and reduced mortality rates in an ischemic rat heart failure model [145]. Moreover, expression of the hepatocyte growth factor (HGF) led to improved heart function in a postinfarct pig heart model [146]. In other approaches, cardiac arrythmia has been treated with Ad vectors expressing Connexin 43 (Cx43) or the I(Kr) potassium channel alpha subunit, resulting in increased conduction velocity, prevention of atrial fibrillation, and reduced tachycardia after myocardial infarction in pigs [147] and prevention of fibrillation in a swine model [148], respectively. The pMX5 retrovirus has been applied for the expression of the transcription factors GATA4, MEF2C, and TBX5 for the reprogramming of non-myocytes in the mouse heart to cardiomyocyte-like cells to reduce infarct size and to attenuate cardiac dysfunction [149].

**Table 3 viruses-15-00698-t003:** Preclinical and clinical examples of viral vectors applied for cardiovascular, metabolic, and hematological diseases.

Viral Vector	Phase	Findings
**Cardiovascular**		
Ad-SERCa2a	Pre	Restoration of systolic/diastolic heart function in rat heart model [142]
Ad-SERCa2a	Pre	Improved coronary blood flow, reduced cardiomyocyte size in rats [143]
AAV1-SERCa2a	Pre	Increased coronary blood flow in pig model [144]
LV-SERCa2a	Pre	Protection against dilation, improved systolic and diastolic functions [145]
Ad-HGF	Pre	Improved heart function in a post-infarct pig model [146]
Ad-Cx43	Pre	Prevention of atrial fibrillation, reduced tachycardia in post-infarct pigs [147]
Ad-KCNH2	Pre	Prevention of fibrillation in swine model [148]
pMX5-GATA4/TBX5	Pre	Reprogramming cells to reduce infarct size, attenuated cardiac dysfunction [149]
Ad-VEGF	Phase I	Improved myocardial perfusion reserve, relief in symptoms in angina patients [150]
Ad-VEGF	Phase II	Improved treadmill exercise, no improvement in myocardial perfusion [151]
Ad-FGF4	Phase I/II	Improved treadmill exercise [152,153], stress-induced myocardial perfusion [154]
AAVI-SERCa2a	Phase I	Improved in functional, symptomatic, ventricular/remodeling parameters [155]
AAV1-SERCa2a	Phase II	Improved walking, oxygen consumption, ventricular endosystolic volume [156]
AAV1-SERCa2a	Phase IIa	Reduced number of cardiovascular events and deaths [157]
**Metabolic**		
AAV-GUS	Pre	Single injection reversed mucopolysaccharidosis phenotype in mice [158]
AAV-LDL-R	Pre	Nearly normal lipid levels, prevention of severe atherosclerosis in mice [159]
AAV-FGF21	Pre	Therapeutic efficacy in transgenic mice as model for T2DM [160]
AAV8-PAL	Pre	Long-term correction of hyperphenylalaninemia in mice [161]
AAV8-GAA	Pre	Therapeutic activity and attenuated Pompe disease phenotype in mice [162]
MSCV-Insulin	Pre	Decreased blood glucose, increased insulin, reversal of diabetes in mice [163]
MMTV-Ad36 E4orf1	Pre	Improved glucose excursion in mice [164]
AAV-PBGD	Phase I	Unable to correct AIP phenotype, but reduced hospitalization [165]
AAV-hAAT	Phase I	Above background levels of hAAT in patients [166]
AAV-hAAT	Phase II	Strong immunostaining of AAT in muscle biopsies [167]
**Hematology**		
Ad-FVIII	Pre	Physiological levels of FVIII in mice [168]
Ad-FIX	Pre	Long-term expression of FIX in nude mice [169]
Ad-cFIX	Pre	Correction of hemophilia B in dogs, but only 1–2% FIX after 3 weeks [170]
Ad-cFIX + CsA	Pre	CsA restored therapeutic FIX levels for at least 6 months [171]
AAV6/AAV8-FVIII	Pre	Therapeutic levels of FVIII lasting for >3 years in dogs [172]
AAV8-FVIII	Pre	1–2% of normal FVIII levels, prevention of 90% of bleeding episodes in dogs [173]
AAV8/AAV9-FVIII	Pre	1.9–11.3% of normal FVIII, no effect on chromosomal integration in dogs [174]
AAV8-FIX	Pre	25–40% of normal FIX levels in hemophilic dogs [175]
AAV-FVIII	Phase I/II	8–60% of normal FVIII levels in hemophilia A patients [176]
AAV5-hFVIII-SQ	Phase I	Clinical benefits, reduced bleeding events in hemophilia A patients [177]
AAV8-FIX	Phase I	1–6% of normal FIX levels in hemophilia B patients for 3.2 years [178]
scAAV2-FIX	Phase I	Stable expression of FIX for 7 years, reduced bleedings in patients [176]
AAVS3-FIX	Phase I/II	Stable expression for 27 months required immunosuppression in patients [179]
AAV5-FVIII	Approval	Conditional marketing approval for severe hemophilia A by EMA [180].
2bF8 LV	Pre	Sustained FVIII expression. correction of hemophilia A phenotype in mice [181]
SIN-LV-cFIX	Pre	Long-term stable expression of FIX in dogs [182]
2bF9/MGMT LV	Pre	2.9-fold increase in FIX expression, reduced blood clotting time [183]
LV-PKDL/R	Pre	LV-transduced HSCs corrected hemolytic anemia phenotype in mice [184]
MSCV-FANCA	CR	Transient gene correction in 2 Fanconi anemia patients [185]
LV-RPS19	Pre	Cure of DBA in an RPS19 DBA-deficient mouse model [186]
LentiGlobin BB305	Phase I	Stop of transfusion of red blood cells in β-thalassemia patients [187]
LentiGlobin BB305	Phase III	Sustained HbA^T87Q^, non-β^0^/β^0^ genotype patients independent of transfusions [188]
GLOBE LV	Pre	In utero gene therapy providing normalized hematological phenotype in mice [189]
GLOBE LV	Phase I/II	Transfusion discontinued or reduced in β-thalassemia patients [190]
LV-HSCs	Pre	Anti-sickling protein expression in mice [191]
LentiGlobin BB305	CR	Transfusions in the SCD patient could be discontinued [192]
LentiGlobin BB305	Phase I/II	Clinical remission or reduced frequency of transfusions in SCD patients [193]
HIV-HSV-TK	Pre	Prolonged survival of mice with acute T-cell leukemia (ATL) [194]
SIN-GALV.fus	Pre	Antitumor activity against acute myeloid leukemia (AML) xenografts in mice [195]
AAV6-CD33-iCasp9	Pre	Antitumor and apoptotic activity, prolonged survival in zebrafish [196]
LOAd703 + CAR T	Pre	Lymphoma killing in cell lines and in xenograft mouse models [197]
HSVrantes/HSVB7.1	Pre	Complete tumor regression after combination therapy in mice [198]
HSV-1 T-01	Pre	Intratumoral and contralateral tumor regression in mice [199]
AAV8-h1567 mAb	Pre	Strong antitumor activity, prolonged survival in mice [200]
SIN + α4-IBB Ab	Pre	Complete lymphoma eradication, long-lasting immunity in mice [201]
CVA21 RNA	Pre	Rapid tumor regression in mice, comparable to CVA21 particles [202]
VSV-IFN-β	Pre	Eradication of tumors, prolonged survival in mice [203]
Reolysin	Pre	Reduced tumor burden in xenograft and syngeneic myeloma mouse models [204]

2bF8 LV, LV vector with integrin alpha-2b promoter; 2bF9/MGMT LV, LV vector with alpha-2b promoter; FVIII gene; hAAV, adeno-associated virus; AAVS3, AAV3 with synthetic capsid protein; Ad, adenovirus; AIP, acute intermittent porphyria; CR, case report; CsA, cyclosporin A; Cx43, connexin 43; DBA, Diamond-Blackfan anemia; EMA, European Medicines Agency; FANCA, Fanconi anemia complementation group A; FGF4, fibroblast growth factor 4; FGF21, fibroblast growth factor 21; FIX, factor IX; FVIII, factor VIII; hAAT, human alpha-1-antitrypsin; GAA, acid α-glucosidase; GALV.fus, gibbon ape leukemia virus fusion protein; GUS, β-glucuronidase; h1567 mAb, anti-CCR4 monoclonal antibody; HbA, hemoglobin; HSCs, hematopoietic stem cells; HSV, herpes simplex virus; HSV-TK, herpes simplex virus-thymidine kinase; KCNH2, I(Kr) potassium channel alpha subunit; LDL-R, low density lipoprotein receptor; LentGlobin BB305, LV vector expressing HbA^T87Q^; MMTV, mouse mammary tumor virus; MSCV, murine stem cell virus; PAL, phenylalanine amino lyase; PBGD, porphobilinogen deaminase; pMX5, retrovirus; Pre, preclinical studies; RPS19, ribosomal protein S19; scAAV8, self-complimentary AAV8; SERCa2a, sarcoplasmic reticulum Ca^2+^ ATPase; SIN-LV, self-inactivating LV; SIN, Sindbis virus; T2DM, type 2 diabetes mellitus; VEGF, vascular endothelial growth factor; VSV, vesicular stomatitis virus.

Related to clinical evaluation, in a phase I trial, intramyocardial administration of the vascular endothelial growth factor (VEGF) expressed from Ad vectors generated improvement in myocardial perfusion reserve and relief of symptoms in refractory angina patients [150]. In a phase II study in patients with severely symptomatic coronary artery disease, the Ad-VEGF vector showed significant improvement in treadmill exercise, although, no improvement in myocardial perfusion was observed [151]. In a series of phase I-II AGENT (Angiogenic GENe Therapy) trials, the fibroblast growth factor 4 (FGF4) was expressed from Ad vectors in patients with chronic stable angina [152,153,154]. The studies demonstrated symptomatic improvement in exercise time [152], sex-specific benefits for treadmill exercise [153], and improvement in stress-induced myocardial perfusion [154]. AAV1-SERCa2a has been evaluated in a phase I study in patients with heart failure, which led to an improvement in functional, symptomatic, and ventricular/remodeling parameters [155]. In a phase II study, improvements in a walking test, peak maximum oxygen consumption, and left ventricular endosystolic volume were seen in patients with class III/IV heart failure after AAV1-SERCa2a treatment [156]. In another phase IIa trial, AAV1-SERCa2a treatment reduced the number of cardiovascular events and deaths [157].

### 3.3. Metabolic Diseases

More than 30 metabolic diseases have been subjected to viral vector-based gene therapy studies [205] (Table 3). AAV vectors have been used in the majority of studies. For example, AAV-based expression of β-glucuronidase (GUS) has been used for treatment of the lysosomal storage disease mucopolysaccharidosis [158]. Intramuscular injection of AAV-GUS generated high levels of local GUS. In contrast, only low GUS activity was detected after intravenous administration in mice [158]. However, even low levels of GUS reduced the glycosaminoglycan levels to normal in the liver and reduced storage granules substantially, and a single administration of AAV-GUS was sufficient to reverse the disease phenotype in mice [158]. AAV vectors have also been used for the expression of the low-density lipoprotein receptor (LDL-R) in the liver, which provided nearly complete normalization of serum lipid levels and prevention of severe atherosclerosis in mice [159]. Related to type 2 diabetes mellitus (T2DM), expression of the fibroblast growth factor 21 (FGF21) from AAV vectors provided substantial reduction in body weight, adipose tissue hypertrophy and inflammation, and insulin resistance for more than one year in transgenic ob/ob mice or wildtype mice receiving a high-fat diet [160]. In the context of phenylketonuria (PKU), a single injection of an AAV8 vector, containing the human antitrypsin (hAAT) promoter for the liver-specific expression of phenylalanine amino lyase (PAL), generated long-term correction of hyperphenylalaninemia in mice [161]. Moreover, AAV8 vectors expressing the acid α-glucosidase (GAA) gene have been evaluated for the treatment of Pompe disease, a glycogen storage disease [162,206]. Liver-specific GAA expression led to therapeutic activity and attenuated the disease phenotype in mice. RVs such as murine stem cell virus (MSCV) have been used for expression of the human insulin gene in diabetic mice, showing decrease in blood glucose levels, increase in secreted insulin, and reversal of diabetes for up to 6 weeks [163]. Moreover, the hyperglycemic Ad36 E4orf1 protein was expressed from an murine mammary tumor virus (MMTV) vector generating improved glucose excursion in C57BL/6 mice despite their high fat diet, and enhanced glucose levels without increasing insulin sensitivity [164].

In the case of clinical trials, intravenous administration of AAV particles expressing the porphobilinogen deaminase (PBGD) gene in a phase I trial in patients with acute intermittent porphyria (AIP) did not correct the AIP phenotype but suggested a trend towards a reduction in hospitalization and heme treatment [165]. In another approach, a phase I trial on patients with alpha-1-antitrypsin (AAT) deficiency was conducted with AAV vectors expressing the human AAT gene [166]. The safe intramuscular administration of AAV-hAAT generated AAT expression above background levels, which was sustained for at least one year. A follow-up phase II trial demonstrated antibody responses in all patients, however, not against AAT [167]. Despite that, strong immunostaining of AAT was detected in muscle biopsies.

### 3.4. Hematological Diseases

Among hematological diseases, hemophilias have been successful targets for gene therapy to correct the mutated factor VIII (FVIII) [207] and factor IX (FIX) [208] genes causing hemophilia A and B, respectively (Table 3). Originally, Ad vectors were applied showing sustained expression of the full-length FVIII at physiological levels in mice [168]. Furthermore, Ad-based long-term FIX expression of more than 300 days could be established in nude mice [169]. Ad-based expression of the canine FIX (cFIX) provided complete correction of the hemophilic phenotype in FIX-deficient hemophilia B dogs [170]. However, the cFIX levels decreased to only 1–2% of normal FIX levels in three weeks, but co-administration of the immunosuppressive cyclosporin A (CsA) restored therapeutic FIX levels and correction of hemophilia B for at least 6 months [171].

The limited packaging capacity of AAV vectors has presented some difficulties related to hemophilia therapy due to the large size of the FVIII gene [209]. For this reason, the B-domain deleted (BDD) FVIII has been expressed from AAV vectors [210]. In addition, the choice of AAV serotype is important as AAV8 provided much higher FVIII activity than AAV2, 3, 5, and 7 serotypes [211]. For example, AAV2-based expression of the canine BDD FVIII was only transient, while AAV6 and AAV8 vectors provided persistent therapeutic levels of FVIII, lasting for more than 3 years [172]. In another canine study on AAV8-FVIII, 1–2% of normal FVIII levels were achieved, which prevented 90% of bleeding episodes [173]. Moreover, a study with AAV8 and AAV9 in nine dogs showed 1.9–11.3% of normal levels monitored for 10 years [174]. Liver samples from six dogs identified 1741 unique integration sites in the genome, none of which induced tumors or altered liver function. Related to hemophilia B, AAV8-based FIX delivery increased FIX expression by 8–12-fold, with 25–40% of normal FIX levels in hemophilic dogs [175].

In clinical trials, interim results from a phase I/II study in six hemophilia A patients treated with a single injection of AAV-FVIII generated 8–60% of normal FVIII levels [176]. Moreover, a single infusion of the AAV-FVIII SQ variant (AAV5-hFVIII-SQ) showed sustained clinically relevant benefits with a decrease in bleeding events, and no need for prophylactic FVIII use in severe hemophilia A patients in a multiyear follow-up study [177]. AAV8-FIX particles were evaluated in a phase I trial in hemophilia B patients, which provided 1–6% of normal FIX levels for at least 3.2 years [178]. In another approach, self-complementary AAV2 vectors expressing FIX (scAAV2-FIX) showed stable FIX production for 7 years, contributing to substantial reduction in bleeding in hemophilia B patients [176]. In a phase I/II study, the AAVS3 vector, containing a synthetic capsid protein, was subjected to expression of FIX (FLT180a), which resulted in dose-dependent increase in FIX levels with five patients showing 51–78%, three patients 23–43%, and one patient 260% of the normal FIX levels [179]. Although sustained FIX expression was detected for 27 months, immunosuppression with glucocorticoids was required in all patients. Approval for conditional marketing of an AAV5 vector expressing the BDD FVIII cDNA for the treatment of severe hemophilia A has been granted by the European Medicines Agency (EMA) [180].

LV vectors have also been evaluated for hemophilia gene therapy. For example, the FVIII gene was expressed from a platelet-specific integrin alpha 2b promoter engineered into an LV vector (2bF8 LV) and transduced into mouse bone marrow [181]. Mice transplanted with 2bF8 LV-transduce bone marrow generated functional FVIII activity, survival of tail clipping, and correction of the hemophilia A phenotype. In the case of hemophilia B, expression of cFIX from a self-inactivating LV (SIN-LV) vector, carrying a hepatocyte-specific promoter, generated long-term stable FIX expression in dogs [182]. In another approach, the 2bF9/MGMT LV vector, which contains the alpha-2b promoter, the FIX, and methylguanine-DNA-methyltransferase (MGMT) 140K genes, provided a 2.9-fold higher FIX expression and 3.7-fold higher FIX activity in platelets after hematopoietic stem cell (HSC) transduction [183]. In transplanted mice, the blood clotting time was significantly reduced while the expression of therapeutic platelet-FIX was enhanced in mice.

Hemolytic anemia has been approached by transduction of HSCs by LV expressing the pyruvate kinase L/R (PKL/R) to compensate for pyruvate kinase deficiency (PKD), which corrected the hematological phenotype in mice [184]. The oncoretroviral MSCV vector has been used for ex vivo transfer of the Fanconi anemia complementation group A (FANCA) gene to treat Fanconi anemia (FA) [185]. Despite good safety and tolerability, the gene correction was transient due to the low dose of infused gene-corrected cells. In the context of Diamond-Blackfan anemia (DBA), LV-based expression of the ribosomal protein S19 (RPS19) provided cure of DBA and lethal bone marrow in an RPS19-deficient DPA mouse model [186].

In addition, β-thalassemia caused by more than 200 mutations in the β-globin gene [212] has been the target for viral-based gene therapy. For example, ex vivo transduced LentiGlobin BB305, an LV vector expressing the adult human hemoglobin T87Q mutant gene (HbA^T87Q^), allowed 12 β-thalassemia patients with the β^0^/β^0^ genotype to stop red blood cell transfusions and in 9 other patients, the transfusion volume could be reduced by 73% in a phase I study [187]. Interim results from a phase III trial with LentiGlobin BB305 confirmed the expression of sustained levels of HbA^T87Q^ and for patients with the non-β^0^/β^0^ genotype to become independent of transfusions [188]. The GLOBE LV vector has been subjected to intrahepatic administration in utero in a humanized mouse model, which resulted in a normalized hematological phenotype at 12–32 weeks of age [189]. In a phase I/II trial, rapid recovery was achieved in three adult and six pediatric β-thalassemia patients treated with GLOBE LV vector-transduced stem cells [190]. The transfusion could be completely discontinued in children and reduced in adults.

In the context of sickle cell disease (SCD), which is caused by a single mutation in the β-globin chain of the adult α2β2 hemoglobin tetramer [213], HSCs transduced with LV vectors expressing a βA-globin variant have demonstrated long-term expression for 10 months and accumulation of anti-sickling protein up to 52% of total hemoglobin in mouse models [191]. In a case report, LentiGlobin BB305-transduced bone marrow cells showed no SCD-related clinical events and the patient’s transfusions could be discontinued [192]. In a phase I/II trial, autologous CD34^+^ cells were transduced with LentiGlobin BB305 expressing the anti-sickling β^A-T87Q^ globin gene, which caused no adverse events in three SCD patients [193]. Clinical remission was observed in two patients, and the frequency of transfusions could be reduced in one patient.

Among hematological diseases, leukemias, lymphomas, and myelomas have also been subjected to gene therapy applications using viral vectors, as described previously in more detail [214]. Briefly, LV (HIV) vectors expressing herpes simplex virus-thymidine kinase (HSV-TK) were administered intraperitoneally to adult T-cell leukemia (ATL)-NOD-SCID mice, which generated significantly lower levels of secreted IL-2 and prolonged survival of mice compared to administration of an HIV vector expressing GFP [194]. Expression of a hyperfusogenic gibbon ape leukemia virus envelope glycoprotein (GALV.fus) from a SIN vector resulted in antitumor activity against human acute myeloid leukemia (AML) xenografts in mice [195]. In another approach, the AAV6-CD33 vector carrying an antibody-binding CD33 epitope targeting leukemia cells was utilized for the expression of the inducible caspase 9 (iCasp9) suicide gene in an AML xenotransplantation model in zebrafish [196]. AAV6-CD33-iCasp9 treatment resulted in antileukemic activity, a higher number of apoptotic cells, and prolonged survival.

In the case of lymphomas, the oncolytic Ad vector LOAd703 expressing CD40L and 4-1BBL was combined with chimeric antigen receptor (CAR) T-cell therapy, demonstrating increased killing of lymphoma cell lines and lymphomas in xenograft mouse models [197]. HSV amplicon vectors have been used for the expression of RANTES (HSVrantes) and the T-cell costimulatory ligand B7.1 (HSVB7.1) [198]. Complete EL4 tumor regression was observed in mice after intratumoral co-administration of HSVrantes and HSVB7.1, and in contralateral tumors. Similarly, intratumoral injection of the third generation HSV-1 T-01 vector provided tumor regression not only in injected tumors but also in non-injected contralateral tumors in mice [199]. In another approach, AAV8 expressing the humanized single-chain variable fragment (scFV)-Fc fusion minibody of the anti-CCR4 monoclonal antibody h1567 showed strong antitumor activity and prolonged survival in mice after a single intravenous infusion [200]. The oncolytic SIN vector combined with the agonistic monoclonal antibody to the T-cell stimulatory receptor 4-1BB (α4-1BB Ab) showed complete eradication of a non-Hodgkin B cell lymphoma in an A20 mouse tumor model, and long-lasting antitumor immunity was established in surviving mice [201].

In the context of lymphomas, infectious oncolytic CVA21 RNA was intratumorally injected into KAS6/1 myeloma-bearing mice leading to rapid tumor regression, which was comparable to injection of fully infectious CVA21 particles [202]. Moreover, intravenous administration of the oncolytic VSV vector expressing interferon-β (IFN-β) eradicated myeloma cells and prolonged survival in immune-competent myeloma mice [203]. In addition, the oncolytic reovirus (Reolysin) showed selective replication and induced apoptosis in multiple myeloma cell lines and reduced the tumor burden in xenograft and syngeneic multiple myeloma mouse models [204].

### 3.5. Neurological Disorders

Several approaches have been explored for gene therapy of neurological disorders (Table 4). For instance, AAV-based expression of the glutamic acid decarboxylase 65 (GAD65) gene improved symptoms related to Parkinson’s disease in a rat model, and relieved pain in a rat pain model [215]. In a comparative study, the glial cell-derived neurotrophic factor (GDNF) was expressed from Ad, AAV, and LV vectors resulting in regionally restricted GDNF expression in the striatum and substantia nigra, inhibition of toxin-induced degeneration of nigral dopamine neurons, and functional striatal dopamine innervation in a rat Parkinson’s disease model [215]. Moreover, administration of AAV-GDNF or LV-GDNF to 6-hydroxydopamine (6-OHDA)-lesioned rats and 1-methyl-4-phenyl-1,2,3,6-tetrahydropyridine (MTTP)-lesioned primates generated sustained GDNF delivery for 3–6 months, which contributed to regeneration and functional recovery [216]. In another study, it was demonstrated that LV-GDNF administration to the striatum and substantia nigra reversed functional and motor deficits and completely prevented nigrostriatal degradation in MPTP-lesioned rhesus macaques [217]. In clinical settings, in a phase I clinical trial, the human aromatic-l-amino acid decarboxylase (hAAD) expressed from an AAV vector showed good tolerance, only minor adverse events, and a significant improvement in the Parkinson’s Disease Rating Scale (UPDRS), which was sustained for at least 2 years in patients with moderate to advanced Parkinson’s disease [218]. In a phase I/II clinical trial, tyrosine hydroxylase (TH), aromatic amino acid dopa decarboxylase (AADC), and GTP-cyclohydroxylase-1 (GCH-1) expressed from LV vectors (ProSavin) were subjected to intrastriatal administration in Parkinson’s disease patients, which was safe, well tolerated, and provided significant improvement of motor function [219]. Moreover, a long-term phase I/II follow-up study with ProSavin showed a significant improvement in the UPDRS score 4 years after the treatment [220].

**Table 4 viruses-15-00698-t004:** Preclinical and clinical examples of viral vectors applied for neurological disorders, muscular diseases, and immunodeficiency.

Viral Vector	Phase	Findings
**Neurological**		
AAV-GAD65	Pre	Improved symptoms of Parkinson’s disease in rats [215]
AAV-GAD65	Pre	Pain relief in rat pain model [215]
Ad-GDNF	Pre	Inhibition of toxin-induced degeneration of neurons in rat model [216]
AAV-GDNF	Pre	Inhibition of toxin-induced degeneration of neurons in rat model [216]
LV-GDNF	Pre	Inhibition of toxin-induced degeneration of neurons in rat model [216]
AAV-GDNF	Pre	GDNF for 3–6 months, regeneration, functional recovery in rats, primates [216]
LV-GDNF	Pre	GDNF for 3–6 months, regeneration, functional recovery in rats, primates [216]
LV-GDNF	Pre	Reversed functional and motor deficits, prevented degradation in primates [217]
AAV-hAAD	Phase I	Significant improvement in UPRDS in Parkinson’s disease patients [218]
LV-ProSavin	Phase I/II	Safe, well tolerated, improved motor function in Parkinson’s disease patients [219]
LV-ProSavin	Phase I/II	Significantly improved 4-year UPRDS score in Parkinson’s disease patients [220]
AAV2/5-NGF	Pre	Long-term neuroprotection in rat Alzheimer’s disease model [221]
AAV2/5-NGF	Phase I	Inconclusive results in Alzheimer’s disease patients [222]
AAV-APPsα	Pre	Functional special memory, mitigated synaptic and cognitive deficits in mice [223]
LV-GDNF	Pre	Preserved learning and memory in mouse Alzheimer’s disease model [224]
LV-Klotho	Pre	Less cognitive deficits and Alzheimer’s disease-like pathologies in mice [225]
AAV5-miHTT	Pre	Prevention of ATT aggregate formation, neuronal dysfunction in HD rat model [226]
AAV-miHTT	Pre	Reduced mutant HTT mRNA and protein in transgenic HD minipig brain [227]
AAV-miHTT	Phase I/II	Study in progress on disease progression in Huntington’s disease patients [228]
AAV9-MeCP2	Pre	Prolonged survival in a mouse Rett syndrome model [229]
AAV9-SMN	Phase I	Improved motor function, prolonged survival in SMA patients [230]
AAV9-SMN	Phase I	Improved motor function, prolonged survival in SMA patients [231]
AAV9-SMN	Approval	Approved for treatment of SMA patients in the US, the EU, and Canada [232]
**Muscular**		
Ad-ΔDys	Pre	Restored dystrophin protein levels in mice [233]
AAV-µDys	Pre	Amelioration of dystrophin phenotype in transgenic mtx mice [234]
AAV6-µDys	Pre	Reduced skeletal muscle pathology, prolonged lifespan in dystrophic mice [235]
AAV6-µDys	Pre	Efficient delivery of dystrophin in canine dystrophin model for 2 years [236]
AAV6-µDys	Phase I/II	Therapeutic levels of µDys, improved NSAA score in all DMD patients [237]
AAV9-µDys	Phase I	Study in progress in 4–12-year-old DMD patients [238]
AAV-PABPN1	Pre	Decreased muscle fibrosis, normal muscle strength in OPMD mouse model [239]
LV-PABPN1	Pre	Efficient ex vivo transduction and rescue of myoblasts from OPMD patients [240]
**Immunodeficiency**		
γRV-IL2RG	CR	Long-lasting clinical benefits in 8 out of 10 SCID-X1 patients [241]
γRV-IL2RG	CR	Normal growth, protection against infections in SCID-X1 patients after 18 years [242]
γRV-IL2RG	CR	Sustained clinical benefits in 10 SCID-X1 patients [243]
γRV-IL2RG	CR	T-ALL in SCID-X1 patients after unfavorable integration of the γRV vector [8,244]
SIN-γRV	CR	Successful treatment of 9 SCID-X1 patients without leukemia development [245]
SIN-LV	CR	Successful treatment of 44 SCID-X1 patients without leukemia development [245]
SIN-LV-ABCD1	CR	Prevention of progressive demyelination, clinical benefits in ALD patients [40]
SIN-γRV/LV-ADA	CR	Sustained ADA expression, metabolic correction in >100 SCID-ADA patients [246]

AAV, adeno-associated virus; ABCD1, adenosine triphosphate.binding cassette transporter; Ad, adenovirus; ADA, adenosine deaminase; ALD, adrenoleukodystrophy; ΔDys, truncated dystrophin; DMD, Duchenne muscular dystrophy; GAD65; glutamic acid decarboxylase; GDNF, glial-derived neurotrophic factor; hAAD, human aromatic-l-amino decarboxylase; HD, Huntington’s disease; IL2RG, interleukin-2 receptor gamma subunit; HTT, huntingtin; LV, lentivirus; MeCP2, methyl CpG binding protein 2; µDys, mircro-dystrophin; miHTT, micro-RNA targeting HTT; NSAA, North Star Ambulatory Assessment; OPMD, oculopharyngeal muscular dystrophy; PABPN1, poly A-binding protein nuclear 1; Pre, preclinical studies; ProSavin, LV vector expressing tyrosine hydroxylase, aromatic amino acid dopa decarboxylase, and GTP-cyclohydroxylase-1; γRV, gamma retrovirus; SCID-X1, X-linked severe combined immunodeficiency; SIN-LV, self-inactivating LV; SMA, spinal muscular atrophy; SMN, survival motor neuron; T-ALL, T-cell acute lymphoblastic leukemia; UPDRS, United Parkinson’s Disease Rating Scale.

In the case of Alzheimer’s disease, a chimeric AAV2/5 vector with the AAV2 genome and the AAV5 capsid structure has been applied for the expression of the nerve growth factor (NGF) [221]. In comparison to AAV2-NGF, the AAV2/5-NGF showed superior transduction of septal cholinergic neurons in rats, which provided long-term neuroprotection. Although preclinical studies have shown promising results regarding neuroprotection, the results from a phase I trial with AAV2/5-NGF were inconclusive [222]. In another approach, the secreted amyloid precursor protein (AAPsα) was expressed from AAV vectors, which resulted in functional rescue of spatial memory and mitigated synaptic and cognitive deficits in mice [223]. Moreover, LV-GDNF administration preserved learning and memory in mice; although, the amyloid and tau pathologies were not reduced [224]. However, the upregulation of the brain-derived neurotrophic factor (BDNF) was induced, which can contribute to neuronal protection against atrophy and degeneration. In another study, LV-based expression of the anti-aging gene Klotho efficiently ameliorated cognitive deficits and Alzheimer’s disease-like pathologies in the brains of APP/presenilin-1 transgenic mice [225].

Huntington’s disease, caused by a mutation in the huntingtin (HTT) gene, has been explored for AAV-based gene silencing with miRNAs targeting HTT [226]. Administration of AAV5-miHTT suppressed mutant HTT mRNA, resulting in almost complete prevention of mutant HTT aggregate formation and suppression of DARPP-32-associated neuronal dysfunction in a rat model for Huntington’s disease [226]. Moreover, AAV5-miHTT significantly decreased mutant HTT mRNA and protein levels in the brain of transgenic HD minipigs [227]. A phase I/II clinical trial is in progress for the evaluation of safety, tolerability, and proof-of-concept of a single-time bilateral injection of AAV-miHTT (AMT-1309) in adults with early-stage Huntington’s disease compared with control individuals for disease progression [228]. In the context of the X-linked Rett syndrome (RTT), the transcription regulator methyl CpG-binding protein 2 (MeCP2) was expressed from an AAV9 vector showing prolonged survival in an RTT mouse model [229]. In attempts to treat spinal muscular atrophy (SMA), which is associated with muscle weakness and atrophy, but caused by deterioration of motor neurons in the brainstem and spinal cord, an AAV9 vector has been employed for the expression of the survival motor neuron (SMN) gene [230]. In a phase I trial, AAV9-SMN delivery generated remarkable improvements in motor function and survival rates [230]. In another phase I study, a single intravenous AAV9-SMN injection improved motor function and extended survival in SMA patients [231]. AAV9-SMN1 has been approved in the US for treatment of children with SMA up to the age of two years, and in the EU and Canada in SMA patients under the brand name Zolgensma [232].

### 3.6. Muscular Diseases

Several gene therapy applications targeting muscular diseases, particularly various muscular dystrophies, have been successful [233]. For example, related to Duchenne muscular dystrophy (DMD), Ad-based expression of a truncated form of dystrophin restored dystrophin-related protein levels in mouse skeletal muscle [234]. The large size of dystrophin has been a major issue for AAV-based expression due to its limited packaging capacity, which has led to the engineering of ”micro-dystrophin” cassettes (µDys) [235]. AAV-µDys were used for the production of transgenic mtx mice, which ameliorated the dystrophin phenotype with restored levels of normal C57BL/10 mice [235]. Moreover, AAV6-µDys restored dystrophin levels in respiratory, cardiac, and limb musculature, reducing the skeletal muscle pathology, and substantially prolonging the lifespan of severely dystrophic mice [236]. Additionally, the AAV6-µDys resulted in efficient delivery of dystrophin throughout different skeletal muscles in a canine dystrophin model, which lasted for at least two years [237]. In the context of clinical trials, AAV6-µDys has been subjected to a phase I/II trial in DMD patients, in which, according to interim results, therapeutic levels of µDys, 81% dystrophin-positive fibers, and improvement in the North Star Ambulatory Assessment (NSAA) score were seen in all patients [238]. Moreover, a phase I trial with the AAV9-mini-dystrophin vector is in progress in 4-12-year-old DMD patients for the verification of safety, tolerability, dystrophin expression and distribution, and muscle strength [247]. Several other AAV-based phase I/II and phase III are in progress in DMD patients (NCT03368742, NCT03375164, and NCT04281485), showing minimal adverse events, good safety in four patients, robust expression of µDys, and functional muscle improvement based on interim results [238].

In the case of oculopharyngeal muscular dystrophy (OPMD), which is caused by trinucleotide repeat expansion in the poly A-binding protein nuclear 1 (PABPN1) gene, patients suffer from late onset of ptosis, swallowing difficulties, and formation of nuclear aggregates in skeletal muscles [239]. Significant reduction in insoluble aggregates, decrease in muscle fibrosis, and normalization of muscle strength was seen in an OPMD mouse model after AAV-PABPN1 administration [240]. For ex vivo studies in myoblasts from OPMD patients, LV-based delivery was utilized due to the low transduction efficacy of AAV in primary myoblasts [241]. In contrast, the LV-PABPN1 transduction was efficient and provided myoblast cell rescue.

### 3.7. Immunodeficiency

The area where gene therapy has seen the greatest progress is undoubtedly in immunodeficiency, and the treatment of SCID and other immunodeficiencies. Despite the great excitement due to successful defective Moloney γRV-based correction of SCID-X1 in children, a major setback was encountered as the therapeutic gene was inserted into the LMO2 proto-oncogene region of the genome leading to leukemia development in a few patients [8,228]. In this first clinical trial, CD34^+^ cells were transduced with the RV vector expressing the interleukin-2 receptor gamma subunit (IL2RG) in 10 SCID patients, which established normal T-cell counts within 3–6 months and demonstrated long-lasting clinical benefits in 8 out of 10 patients [242]. Remarkably, in a follow-up study of 18 years, all but one patient presented normal growth and protection against infections associated with SCID-X1 disease [243]. In another study, sustained clinical benefits were obtained in 10 SCID-X1 patients [244], although 2-14 years after the therapeutic intervention, T-cell acute lymphoblastic leukemia (T-ALL) was discovered in patients where the γRV vector was integrated either into the LMO2 [8] or the CCDN2 locus [245]. For this reason, SIN-γRV vectors have been engineered, which has confirmed that no cases of leukemia developed in nine newly treated SCID-X1 patients [40]. Similarly, engineering of SIN-LV vectors allowed successful treatment of another 44 SCID-X1 patients without any leukemia development [40].

In the context of X-linked adrenoleukodystrophy (ALD), SIN-LV vector expressing the adenosine triphosphate-binding cassette transporter (ABCD1) were ex vivo transduced into patient-derived autologous CD34^+^ cells [248]. When SIN-LV-ABCD1 transduced cells were reinfused in two ALD patients, progressive cerebral demyelination was prevented providing clear clinical benefits [248]. Related to adenosine deaminase-severe combined immunodeficiency (ADA-SCID), the defected adenosine deaminase (ADA) gene [246] has been replaced by delivery with SIN-γRV or SIN-LV vectors [246]. Today, more than 100 ADA-SCID patients have been treated, resulting in sustained ADA expression, metabolic correction, and high overall survival [249].

### 3.8. Other Diseases

In addition to the disease indications described above, other disease areas such as ophthalmologic and lung diseases have been subjected to gene therapy. Moreover, infectious diseases have been mainly subjected to vaccine development, which in a broad sense can be considered as gene therapy. As these areas have previously been described in detail elsewhere [250], only a short summary is included here (Table 5).

Ophthalmology has been considered as a favorable area for gene therapy due to the relatively easy access to treatable space, allowing topical administration of gene therapy vectors. For example, intravitreal administration AAV vectors expressing the brain-derived neurotrophic factor (BDNF) showed protection of retinal ganglion cells, and reduced the intraocular pressure in a rat glaucoma model [251]. The intraocular pressure could also be reduced in mice by overexpression of matrix metalloproteinase 3 (MMP-3) from an AAV vector [252]. Regarding macular dystrophy X-linked retinoschisis (XLRS), the loss of the extracellular matrix protein retinoschisis 1 (RS1) was compensated by AAV-based delivery of RS1 to the eye of RS1 knockout mice, which generated significant improvement in retinal structure and function [253]. AAV vectors have been utilized for gene therapy of color blindness, achromatopsia [254]. As mutations in the cyclic nucleotide gated channel (CNGC) and the guanine nucleotide α-transducin (GNAT) genes cause achromatopsia, AAV-GNAT2 expression under the control of a human red cone opsin promoter was used to restore color vision in mice [255]. Improved photopic electrophysiological responses and functional vision were obtained in dogs subjected to subretinal injection of AAV5-CNGB3 [256].

**Table 5 viruses-15-00698-t005:** Preclinical and clinical examples of viral vectors applied for ophthalmologic and lung diseases, and vaccine development against infectious diseases.

Viral Vector	Phase	Findings
**Ophthalmologic**		
AAV-BDNF	Pre	Retinal ganglion cell protection, reduced intraocular pressure in rat glaucoma [251]
AAV-MMP-3	Pre	Reduced intraocular pressure in mice [252]
AAV-RS1	Pre	Significant improvement in retinal structure, function in RS1 knockout mice [253]
AAV-GNAT	Pre	Restoration of color vision in mice [255]
AAV5-CNGB3	Pre	Improved photopic electrophysiological responses, functional vision in dogs [256]
AAV-sFLT01	Phase I	Good safety and tolerability in AMD patients [257]
AAV-sFLT01	Phase IIa	No serious adverse events, improved vision in AMD patients [258]
AAV2-ND4	Phase I	Significant improvement of visual acuity in LHON patients [259]
AAV2-ND4	Phase I	Enhanced visual acuity in LHON patients [260]
AAV2-RPE65	Phase III	Maximum vision improvement in patients with inherited retinal dystrophy [261]
AAV2-RPE65	Approval	Approved for treatment of visual loss in the US, Australia, and Canada [262]
**Lung**		
AAV-CFTR	Pre	Long-term (6 months) CFTR expression in rabbit airway epithelium [263]
AAV-CFTR	Pre	Safe delivery of CFTR DNA to rhesus macaque lung [264]
HD-Ad-CFTR	Pre	Transduction of airway basal cells from CF patients, restored CFTR activity [265]
HIV-CFTR	Pre	Partial recovery of CFTR function in CF knockout mice for 110 days [266]
FIV-CFTR	Pre	Restored CFTR activity in CF pigs [267]
SIV-CFTR	Pre	Functional CFTR in mouse lung, human air–liquid interface cultures [268]
**Infectious**		
ChAdOx1 nCoV-19	Phase III	Good safety and 62–90% vaccine efficacy [269]
ChAdOx1 nCoV-19	Approval	Granted EUA in the UK [270]
Ad5.S-nb2	Phase II	Strong immunogenicity, good safety in adults [271]
Ad5.S-nb2	Approval	Granted EUA in China [270]
rAd26-S/rAd5-S	Phase III	91.6% vaccine efficacy from interim results [272]
rAd26-S/rAd5-S	Approval	Granted EUA in Russia [273]
Ad26.COV2.S	Phase III	Vaccine efficacy after single dose [274]
Ad26.COV2.S	Approval	Granted EUA in the US [270]
VSV-ZEBOV	Phase III	Good vaccine efficacy in Guinea and Sierra Leone [275,276]
VSV-ZEBOV	Approval	Approval as Ervebo for vaccination against EVD [277]

AAV, adeno-associated virus; Ad, adenovirus; AMD, age-related macular degeneration; BDNF, brain-derived neurotrophic factor; CF, cystic fibrosis; CFTR, cystic fibrosis transmembrane conductance regulator; CNGB3, cyclic nucleotide gated channel B3; EUA, emergency use authorization; FIV, feline immunodeficiency virus; GNAT, guanine nucleotide transducing; HD-Ad, helper-dependent adenovirus; HIV, human immunodeficiency virus; LHON, Leber’s hereditary optic neuropathy; MMP-3, matrix metalloproteinase 3; ND4, NADH dehydrogenase protein subunit 4; Pre, preclinical studies; RS1, retinoschisis 1; sFLOT01, fusion protein of VEGF and the Fc portion of the human IgG1; SIV, simian immunodeficiency virus.

Regarding clinical applications, AAV vectors expressing the sFLT01 fusion protein comprising the VEGF and the Fc portion of the human IgG1 showed good safety and tolerability in a phase I trial in 19 age-related macular degeneration (AMD) patients [257]. Furthermore, no treatment-related serious adverse events were recorded, but improved vision was registered in 11 AMD patients treated with AAV-sFLT01 in a phase IIa study [243,258]. Leber’s hereditary optic neuropathy (LHON), characterized by rapid loss of vision, is caused by a mutation in the NADH dehydrogenase protein subunit 4 (ND4) [278]. AAV2-ND4 treatment resulted in significant improvement in visual acuity in six out of nine LHON patients [259]. In another phase I trial, modest but statistically significant improved visual acuity was seen for 14 LHON patients [260]. Moreover, patients with RPE65-mediated inherited retinal dystrophy were subjected to AAV2-based RPE65 gene replacement therapy in a phase III study, which provided maximum possible vision improvement [261]. AAV2-RPE65, Voretigene neparvovec, has been approved for treatment of visual loss due to inherited retinal dystrophy in patients in the US, Australia, and Canada under the brand name Luxturna [262].

Gene therapy for lung diseases has mainly focused on the potential of developing some breakthrough treatment for cystic fibrosis. As cystic fibrosis is caused by mutations in the cystic fibrosis transmembrane conductance regulator (CFTR) gene [263], viral vector based CFTR expression represents an attractive approach. For example, AAV2-CFTR administered via fiberoptic bronchoscopy to the rabbit lung provided CFTR expression for at least 6 months in the airway epithelium [279]. In another approach, AAV2-CFTR was administered to the right lower lung lobe of rhesus macaques resulting in safe long-term delivery of CFTR DNA [264]. A helper-dependent Ad (HD-Ad) vector has also been engineered for intranasal delivery in mice and bronchoscopic instillation in pigs [265]. The HD-Ad-CFTR also demonstrated transduction of human airway basal cells from cystic fibrosis patients and restoration of CFTR channel activity [265]. Among LV vectors, HIV-based expression of CFTR in the mouse epithelium resulted in a partial recovery of electrophysiological functions in cystic fibrosis knockout mice for at least 110 days [266]. Moreover, a FIV-CFTR based vector pseudotyped with the GP64 protein restored CFTR activity in pigs with cystic fibrosis [267]. SIV-based functional expression of CFTR was also established in mouse lung and in human air–liquid interface cultures as a preparation for the first in-human trial [268].

Finally, vaccine development against infectious diseases using viral vectors has been very successful, recently. Needless to say, the unprecedented rapid development of different Ad-based COVID-19 vaccines has strongly contributed to the downgrading of the COVID-19 pandemic to an endemic status. The ChAdOx1 nCoV-19 vaccine [269], based on the ChAdOx1 chimpanzee Ad, and the Ad5-S-nb2 vaccine [271], based on the human Ad5 serotype, carry the SARS-CoV-2 spike (S) protein as an antigen and have demonstrated high vaccine efficacy in phase III clinical trials after two immunization doses. In contrast, the rAd26-S/dAd5-S (Sputnik) vaccine [272] is based on a prime vaccination with the Ad26 serotype expressing the S protein, followed by a booster vaccination with the Ad5 serotype also expressing the S protein, showing good efficacy in phase II and III studies. The strategy of this vaccination regimen is to limit immune reactions against Ad and reduction in vaccine efficacy by using another Ad serotype for the booster vaccination. The Ad26 serotype-based Ad26.COV2.S vaccine [274] also expresses the S protein, but in contrast to the other Ad-based vaccines, a single immunization has shown efficacy in clinical trials. The positive results from clinical trials supported the granting of Emergency Use Authorization (EUA) for the ChAdOx1 nCoV-19 vaccine in the UK in December 2020, the Ad26.COV2.S vaccine in the US in February 2021, and the Ad5-S-nb2 vaccine in China in February 2021 [270]. Controversially, the rAd26-S/rAd5-S vaccine received approval in Russia already in August 2020, after only being preliminary evaluated in 76 Russian volunteers [273]. Although good safety and vaccine efficacy have been achieved, emerging SARS-CoV-2 variants and detection of rare serious adverse events due to mass vaccinations will require intelligent re-engineering of existing vaccines to meet the new demands.

In the context of other vaccines, the VSV-based Ebola virus vaccine (VSV-ZEBOV) has demonstrated good safety profiles and excellent efficacy in two phase III studies conducted in Guinea [275], and in Guinea and Sierra Leone [276]. In 2020, the VSV-ZEBOV vaccine was approved under the name Ervebo for vaccinations against Ebola virus disease (EVD) [277].

## 4. Conclusions

In summary, viral vectors have been successfully applied for a broad spectrum of disease indications. Encouraging results have been obtained in preclinical studies in animal models, and also in clinical trials in patient groups with long-lasting cure, confirmed especially in SCID-X1 patients [243]. Gendicine^TM^, a replication-deficient Ad vector expressing the p53 gene, was approved in China [280], and more than 30,000 patients with head and neck cancer have been treated with it. Gendicine^TM^ has demonstrated good safety and efficacy, especially in combination with chemo- and radiotherapy [281]. In the US and Europe, a second-generation oncolytic HSV vector expressing GM-CSF has been approved for melanoma therapy [124]. The AAV-based Onasemnogene aboparvovec (Zolgensma) has been approved for the treatment of SMA [232]. Furthermore, approval for the AAV-based Voretigene neparvovec (Luxturna) was received for the treatment of inherited retinal dystrophy [262]. As mentioned above, several Ad-based COVID-19 vaccines have been granted EUA [272], and the Ebola vaccine Ervebo has been approved [277]. However, it is important to keep in mind the case of Glybera^TM^, the AAV-based treatment of lipoprotein lipase deficiency [282]. Despite its approval in Europe, the clinical use of Glybera^TM^ was discontinued due to lack of demand for this rare monogenic inherited disease.

Moreover, several issues need still to be addressed to make viral vector-based gene therapy highly attractive. Despite the advantage of viral vector-based gene delivery compared to non-viral vector systems, the safety of using particularly oncolytic and replication-proficient vectors is of utmost importance. Safety issues have also surfaced related chromosomal integration, where random integration has caused severe adverse events. Another issue of concern has been the difficulties in transferring successful proof-of-concept findings from rodents to larger animals and especially to humans. A potential “bridge” to success, particularly in the field of cancer therapy, has been to target domestic animals. For example, canines develop natural tumors and in addition to developing veterinary drugs, they serve as a potential model for pre-evaluation of efficacy before conducting human trials, partly because they represent good models for delivery to a larger organism, and partly because the natural tumors in canines closely resemble human cancers in contrast to induced and implanted tumors in rodent models.

Furthermore, an often-asked question is which viral vector system, and which therapeutic target should be chosen. Based on all gene therapy examples described in this review, it is obvious that there is not a single vector suitable for all applications. For this reason, viral vector diversity is important in research and development of promoting gene therapy. Due to the extensive number of preclinical and clinical trials conducted with viral vectors, the goal has been to give an overview of which viral vectors are suitable for which indication. For example, self-replicating RNA viruses have proven excellent for high-level short-term transgene expression required for cancer therapy, and development of vaccines against infectious diseases and cancers. In contrast, inherited diseases and chronic diseases, such as immunodeficiency, hematological diseases, and muscular dystrophy, which require long-term expression of therapeutic genes albeit not necessarily at high levels, have favored the application of Ad, AAV, HSV, RV, and LV vectors. Both vectors providing extrachromosomal expression and chromosomal integration have proven useful for therapeutic efficacy, lasting for several years. As with any other method of drug development, the management of serious adverse events is important. Not unexpectedly, the delivery of viral vectors causes adverse events, as does generally any drug. For this reason, efforts have been made to reduce the risk of using viral vectors and to decrease the severity of adverse events by the deletion of non-essential genetic material from viral vectors, the use of attenuated or less cytopathogenic viral vectors, and monitoring the spread of viruses and establishing a control of their replication and expression capacity. As seen for long-term follow-up studies, treatments for several years have not revealed adverse events, including the extreme example of 18 years of therapeutic efficacy without any side effects in SCID-X1 patients treated with RV vectors. These positive findings have encouraged the transition to clinical applications. However, in the light of the ever-tightening requirements associated with clinical evaluation, it is important to, already at an early stage of vector development, include appropriate design and engineering steps to fully comply with the requirements for clinical studies, and to facilitate regulatory implementations.

## Figures and Tables

**Figure 1 viruses-15-00698-f001:**
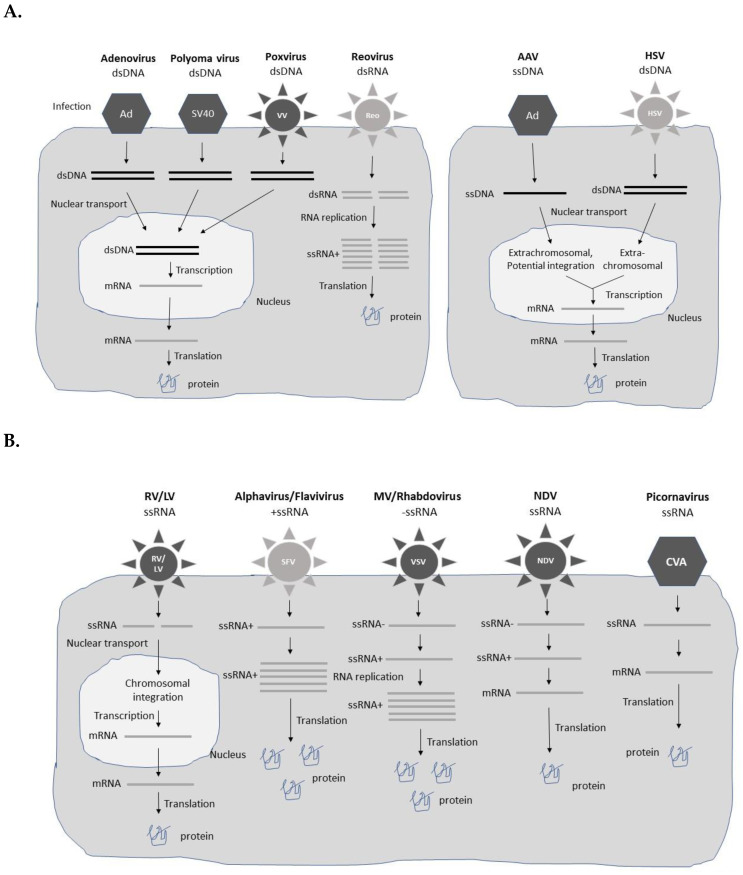
Viral vector expression systems. (**A**) Expression systems engineered for adenoviruses (Ad), simian virus 40 (SV40), vaccinia virus (VV), reoviruses, adeno-associated viruses (AAV), and herpes simplex viruses (HSV). (**B**) Viral vector systems for retro-and lentiviruses (RV), Semliki Forest virus (SFV), measles viruses and vesicular stomatitis virus (VSV), Newcastle disease virus (NDV), and coxsackieviruses A (CVA). dsDNA, double-stranded DNA, dsRNA, double-stranded RNA, ssDNA, single-stranded, DNA, ssRNA, single-stranded RNA, ssRNA+, ssRNA of positive polarity, ssRNA-, ssRNA of negative polarity.

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
