# Peer review of "Viral Vectors in Gene Therapy: Where Do We Stand in 2023?"

_viruses, 2023, doi:10.3390/v15030698_

Round 1
Reviewer 1 Report
This review tries to cover a broad topic (viral vectors for gene therapy). This is a challenging objective, especially for a single author. This kind of review requires careful organization of concepts, which is lacking in the current manuscript. The text is a dense enumeration of articles involving different vectors and transgenes, including a short description of the outcomes. However, little critical insight or interpretation in the context of the respective fields is provided. It is not clear which criteria have been used to select the cited examples among the vast literature.
Other limitation of the manuscript is the high number of inaccuracies (below is only a non-comprehensive list). I recommend that the author seeks advice from experts in specific fields.
Specific comments.
1. The author should make special mention to vectors already approved as treatment for different diseases. Some of them are cited in the conclusions, but not in the respective sections or the tables. For instance, the oncolytic HSV T-VEC for melanoma, AAVs for hemophilia or SMA, etc. Again, it is not clear which criteria have been used to select the cited examples.
2. In the same vein, the abstract mentions some remarkable applications of viral vectors, but not others which are equally relevant.
3. Gene therapy is not defined as “replacement” of genes. It can be supplementation, correction or genes or modulation of gene expression but it is not usually based on gene replacement.
4. The idea that gene therapy has been “dormant until recent years” is just an opinion.
5. Figure 1. HSV does not lead to active chromosomal integration.
6. AAVs are not integrative vectors. Only the wild type virus carries out active integration of their genome in a specific chromosomal location.
7. Transgene expression can persist for several years using non-integrative vectors, provided the organs have a slow cellular turnover.
8. Adenoviral vector versions should be better described. We have oncolytic adenoviruses (replication-competent) on one side, and replication-deficient adenoviral vectors on the other side. Among them, there are E1/E3-deleted (first generation); second generation; and third generation. The latter are also known as helper-dependent, high-capacity or “gutless” adenoviral vectors. Second-generation adenoviral vectors are not “gutless”. All adenoviral vectors but third generation lead to transient expression of transgenes in vivo. In
9. High-Capacity adenoviral vectors and HSV amplicons have a cloning capacity up to 36 and 150 Kb, respectively.
10. The cloning capacity of standard alphaviruses does not exceed 5 Kb.
11. The vector features listed in table 1 are not carefully chosen. For instance, lentiviruses are described as having good cloning capacity (8 Kb), but this is not mentioned in other vectors with 8 Kb capacity, including retroviruses.
12. sFLT01 is a fusion of the VEGF binding domain and a Fc from IgG1. With anti-angiogenic effects.
13. Gendicine is a replication-deficient adenoviral vector. It has antitumor effect, but it is not an oncolytic adenovirus by definition.
14. Line 478. “AAV8 provided much higher FVIII activity than AAV2, 3, 5, 7, and 8 serotypes”. Obviously not higher than AAV8.
Line 495. Self-complementary instead of self-complimentary.
Author Response
This review tries to cover a broad topic (viral vectors for gene therapy). This is a challenging objective, especially for a single author. This kind of review requires careful organization of concepts, which is lacking in the current manuscript. The text is a dense enumeration of articles involving different vectors and transgenes, including a short description of the outcomes. However, little critical insight or interpretation in the context of the respective fields is provided. It is not clear which criteria have been used to select the cited examples among the vast literature.
Other limitation of the manuscript is the high number of inaccuracies (below is only a non-comprehensive list). I recommend that the author seeks advice from experts in specific fields.
Response: Both the Introduction and Conclusion sections have been expanded to add more information and explain the criteria for selection of examples. I also sincerely apologize for inaccuracies and grammatical and spelling mistakes, but the timing of the submission coincided several other urgent tasks. The manuscript has now been thoroughly revisited and revised.
Specific comments.
- The author should make special mention to vectors already approved as treatment for different diseases. Some of them are cited in the conclusions, but not in the respective sections or the tables. For instance, the oncolytic HSV T-VEC for melanoma, AAVs for hemophilia or SMA, etc. Again, it is not clear which criteria have been used to select the cited examples.
Response: Both HSV T-VEC and AAV for hemophilia and SMA are now described in the respective sections of disease.
- In the same vein, the abstract mentions some remarkable applications of viral vectors, but not others which are equally relevant.
Response: As there is a limitation in the word count of the Abstract, only examples could be listed. This is now mentioned in the text and the example of HSV T-VEC approval for melanoma treatment has been added.
- Gene therapy is not defined as “replacement” of genes. It can be supplementation, correction or genes or modulation of gene expression but it is not usually based on gene replacement.
Response: One definition on gene therapy states: “the introduction of normal genes into cells in place of missing or defective ones in order to correct genetic disorders”, which indeed is “replacement” of malfunctional genes. Moreover, the text continues with a broader definition of gene therapy.
- The idea that gene therapy has been “dormant until recent years” is just an opinion.
Response: This is not just “an opinion”, but factual information as major players in biotechnology and drug development distanced themselves from gene therapy activities. Luckily, hard core gene therapists continued to engineer safer and more efficient gene therapy vectors (which is stated in the following sentence!), which contributed to the more recent favorable view on gene therapy as a field. Anyway, “dormant” has been replaced by “put on hold”, which I hope the reviewer will find acceptable.
- Figure 1. HSV does not lead to active chromosomal integration.
Response: My apologies for this mistake! It has now been corrected (also for AAV)
- AAVs are not integrative vectors. Only the wild type virus carries out active integration of their genome in a specific chromosomal location.
Response: I respectfully disagree with the statement that AAV vectors do not integrate into the genome. In fact, approaches have been made to enhance the integration process (see Wang et al. 2012 Mol. Ther. 20, 1912-1911). This point has been included in section 2.2. on AAV.
- Transgene expression can persist for several years using non-integrative vectors, provided the organs have a slow cellular turnover.
Response: This is mentioned for Ad-based expression (last sentence in section 2.1.).
- Adenoviral vector versions should be better described. We have oncolytic adenoviruses (replication-competent) on one side, and replication-deficient adenoviral vectors on the other side. Among them, there are E1/E3-deleted (first generation); second generation; and third generation. The latter are also known as helper-dependent, high-capacity or “gutless” adenoviral vectors. Second-generation adenoviral vectors are not “gutless”. All adenoviral vectors but third generation lead to transient expression of transgenes in vivo. In
Response: Text has been added to the section on adenoviruses (also to Table 1) including descriptions of different adenovirus vectors, including high-capacity adenoviruses and oncolytic adenoviruses.
- High-Capacity adenoviral vectors and HSV amplicons have a cloning capacity up to 36 and 150 Kb, respectively.
Response: This point has been addressed in the text and Table 1.
- The cloning capacity of standard alphaviruses does not exceed 5 Kb.
Response: I respectfully disagree and can give tens of examples from personal experience over 30 years of working with alphaviruses!
- The vector features listed in table 1 are not carefully chosen. For instance, lentiviruses are described as having good cloning capacity (8 Kb), but this is not mentioned in other vectors with 8 Kb capacity, including retroviruses.
Response: Data in Table 1 have been revised accordingly.
- sFLT01 is a fusion of the VEGF binding domain and a Fc from IgG1. With anti-angiogenic effects.
Response: Corrected both in the text and footnote of Table 5. Thank you for pointing it out!
- Gendicine is a replication-deficient adenoviral vector. It has antitumor effect, but it is not an oncolytic adenovirus by definition.
Response: This has been corrected.
- Line 478. “AAV8 provided much higher FVIII activity than AAV2, 3, 5, 7, and 8 serotypes”. Obviously not higher than AAV8.
Response: This has been corrected.
Line 495. Self-complementary instead of self-complimentary.
Response: Corrected
Reviewer 2 Report
1. For table 1, the authors are requested to remove the findings column. Instead, advantages and limitations would be more value added.
2. For all the preclinical and clinical examples, it would be helpful if the authors added the clinical trial number, sponsor, and the year.
3. The authors are also requested to include information about commercial examples of viral based gene therapies eg: Zolgensma, Luxturna.
4. The authors are requested to add more context to the introduction. Why are viral vectors advantageous as compared to other gene delivery vehicles, what are the benefits which would enable their transition into therapeutic applications.
5. The authors are also asked to add a perspective section where they put into perspective where viral vectors are headed in terms of clinical transition, the regulatory outlook, etc.
Author Response
Reviewer 2
For table 1, the authors are requested to remove the findings column. Instead, advantages and limitations would be more value added.
Response: The title of the column has been changed to “Advantages and Limitations”
- For all the preclinical and clinical examples, it would be helpful if the authors added the clinical trial number, sponsor, and the year.
Response: I respectfully disagree as the tables become very busy with additional information. In fact, I actually looked into retrieving the clinical trail numbers, which were neither available in the references cited nor on the clinicaltrials.gov website.
- The authors are also requested to include information about commercial examples of viral based gene therapies eg: Zolgensma, Luxturna.
Response: Descriptions on Zolgensma and Luxturna have been added both to the text and tables.
- The authors are requested to add more context to the introduction. Why are viral vectors advantageous as compared to other gene delivery vehicles, what are the benefits which would enable their transition into therapeutic applications.
Response: Text has been added as suggested by the reviewer.
- The authors are also asked to add a perspective section where they put into perspective where viral vectors are headed in terms of clinical transition, the regulatory outlook, etc.
Response: A section has been added to the Conclusion
Reviewer 3 Report
In the submitted manuscript “Viral Vectors in Gene Therapy: Where Do We Stand in 2023?”, the author gives a comprehensive overview of various viral vector systems and their utilization in preclinical gene therapy studies and clinical trials. First, different viral vector families are discussed regarding their biology, cargo capacity and advantages regarding gene therapy application. The different viral vector expression systems and their findings are summarized in a comprehensive graphic and a well-structured table. In the second part of the review, the author gives examples of how these vector systems can be applied in a gene therapy setting to treat cancer, cardiovascular diseases, metabolic diseases, hematological diseases, neurological disorders, muscular diseases, immunodeficiencies, and other diseases. The findings of pre-clinical studies and clinical trials are again summarized in well-structured tables.
The author tries to include as many studies as possible but states that due to the many ongoing gene therapies, it is impossible to cover all. The reader clearly gets a broad overview of what has been done in the field. In conclusion, the author shortly discusses authorized therapies, their pros and cons and how the future might look like. Overall, the review is well-written and covers a lot of information. Adding some discussion addressing adverse events related to gene therapy, risk assessment and management for safer gene therapies would be helpful. Understandably given the breadth of the review, the information for some viral vector groups appears outdated. Another problem for the un-initiated reader is that it is challenging to figure out the real relevance of the different systems in translational research. This could be addressed by adding a small paragraph in the introduction or conclusion.
Some additional comments:
Adding somewhere in the text that non-integrating and non-replicating vector systems can mediate long-term expression only in post-mitotic tissues would make sense. In proliferating cells, they only transiently persist, which is why applications based on HSCs always need retroviral systems. Due to this unique feature and the special area where RVs are needed it may also make sense to place “retroviruses” separated from all other systems either to the very beginning or the very end of the list.
· Line 20– there is no approved gene therapy for X-SCID to my knowledge. Presumably, the author refers to ADA-SCID?
· Line 76: a word is missing “for at least….”.
· Line 82 – “improved” instead of “approved”
· I think Line 89/90 is misleading in the context of gene therapy. Integration requires a viral factor that is deleted in gene therapy vectors. In practice, AAVs are used to target post-mitotic cells where the vector genomes can persist without integration, such as retinal cells or inner ear hair cells.
· Line 94 – per definition, when HSV is in its latent state it does not support efficient gene expression.
· Line 116 and table 1. Please modify “Retrovirus” to gamma-retrovirus or onco-retrovirus to avoid confusion between family/subfamily names. Eg lentivirus is also a retrovirus.
· Line 122/123 –The cited review is 23 years old and does not reflect what is happening in the field. I suggest deleting the sentence about targeted integration with RV – was never used in GT applications.
· Line 124/125 is also outdated/incorrect – see PMID: 34647983
· Line 134 -139– the term “semi-random” is used. The overall safety profile is similar to SIN-gRV. Adverse events and oncogene integration have been reported in several cases by now (SCD and ALD). The sentence about targeted integration is incorrect – the semi-random integration site distribution was modified, but it is still semi-random. Please update, rephrase.
· Line 140 – Yes LV-producer lines have been described in research papers, but for several reasons (lower titers, residual toxicity of viral vector components) were never used for large-scale production in clinical trials.
· Line 809/810 – please correct the sentence
· Line 219: Is there a reason for the asterisk (*)?
· Table 1 Rhabdovirus: What does “Z7hb gz6u” refer to?
· Table 2: Please explain abbreviation PR, SD, PD in the legend.
· Line 389: One “in” too much in the sentence.
· Line 569: in mice after “a” …
· Table 3: Ad-VEGF report first on Phase I and then on Phase II trial
· Maybe move “Immunodeficiency” paragraph behind the “Hematological diseases” as they both affect the hematopoietic system.
· Line 636: Delete “)” behind miRNAs
· Table 5: Replace “1” with “I”
· Line 794: Delete “)”
· Line 810: another typo?
· Line 821: Delete “the” before highly attractive
Author Response
Reviewer 3
In the submitted manuscript “Viral Vectors in Gene Therapy: Where Do We Stand in 2023?”, the author gives a comprehensive overview of various viral vector systems and their utilization in preclinical gene therapy studies and clinical trials. First, different viral vector families are discussed regarding their biology, cargo capacity and advantages regarding gene therapy application. The different viral vector expression systems and their findings are summarized in a comprehensive graphic and a well-structured table. In the second part of the review, the author gives examples of how these vector systems can be applied in a gene therapy setting to treat cancer, cardiovascular diseases, metabolic diseases, hematological diseases, neurological disorders, muscular diseases, immunodeficiencies, and other diseases. The findings of pre-clinical studies and clinical trials are again summarized in well-structured tables.
The author tries to include as many studies as possible but states that due to the many ongoing gene therapies, it is impossible to cover all. The reader clearly gets a broad overview of what has been done in the field. In conclusion, the author shortly discusses authorized therapies, their pros and cons and how the future might look like. Overall, the review is well-written and covers a lot of information. Adding some discussion addressing adverse events related to gene therapy, risk assessment and management for safer gene therapies would be helpful. Understandably given the breadth of the review, the information for some viral vector groups appears outdated. Another problem for the un-initiated reader is that it is challenging to figure out the real relevance of the different systems in translational research. This could be addressed by adding a small paragraph in the introduction or conclusion.
Response: The Introduction section has been expanded. Discussion has been added to the Conclusion section discussing safer vectors, adverse events, translational research etc.
Some additional comments:
Adding somewhere in the text that non-integrating and non-replicating vector systems can mediate long-term expression only in post-mitotic tissues would make sense. In proliferating cells, they only transiently persist, which is why applications based on HSCs always need retroviral systems.
Response: Information about long-term expression from non-integrating and replication-deficient vectors has been added to section 2.
Due to this unique feature and the special area where RVs are needed it may also make sense to place “retroviruses” separated from all other systems either to the very beginning or the very end of the list.
Response: Although I understand the point made by the reviewer, I think it would be too complicated to separate retrovirus vectors from the other viral vector systems in Tables 2-5.
- Line 20– there is no approved gene therapy for X-SCID to my knowledge. Presumably, the author refers to ADA-SCID?
Response: Corrected
- Line 76: a word is missing “for at least….”.
Response: Corrected
- Line 82 – “improved” instead of “approved”
Response: Corrected
- I think Line 89/90 is misleading in the context of gene therapy. Integration requires a viral factor that is deleted in gene therapy vectors. In practice, AAVs are used to target post-mitotic cells where the vector genomes can persist without integration, such as retinal cells or inner ear hair cells.
Response: This part has been revised. Although AAV vectors generally remain extrachromosomal as the reviewer correctly points out, efforts to develop integrating vectors have been developed, which has also been added to the text and cited in the references.
- Line 94 – per definition, when HSV is in its latent state it does not support efficient gene expression.
Response: This has now been revised accordingly.
- Line 116 and table 1. Please modify “Retrovirus” to gamma-retrovirus or onco-retrovirus to avoid confusion between family/subfamily names. Eg lentivirus is also a retrovirus.
Response: “Retrovirus” has been revised to “γ-Retrovirus” in Table 1.
- Line 122/123 –The cited review is 23 years old and does not reflect what is happening in the field. I suggest deleting the sentence about targeted integration with RV – was never used in GT applications.
Response: I fully agree, the sentence and reference have been removed.
- Line 124/125 is also outdated/incorrect – see PMID: 34647983
Response: I respectfully disagree with the comment. The publication the reviewer refers to focuses on ADA-SCID, whereas the study described (now Ref 39) deals with SCID-X1. Anyway, additional text and references have been added to clarify the differences in insertional oncogenesis seen between ADA-SCID and SCID-X1.
- Line 134 -139– the term “semi-random” is used. The overall safety profile is similar to SIN-gRV. Adverse events and oncogene integration have been reported in several cases by now (SCD and ALD). The sentence about targeted integration is incorrect – the semi-random integration site distribution was modified, but it is still semi-random. Please update, rephrase.
Response: The text has been revised pointing out the “semi-random” integration. Moreover, “targeted integration” has been changed to “modification of integration”.....which reduced the number of integration events”.
- Line 140 – Yes LV-producer lines have been described in research papers, but for several reasons (lower titers, residual toxicity of viral vector components) were never used for large-scale production in clinical trials.
Response: The point mentioned by the reviewer has been included in the text.
- Line 809/810 – please correct the sentence
Response: Corrected
- Line 219: Is there a reason for the asterisk (*)?
Response: Corrected
- Table 1 Rhabdovirus: What does “Z7hb gz6u” refer to?
Response: Corrected, no idea how this was incorporated in Table 1.
- Table 2: Please explain abbreviation PR, SD, PD in the legend.
Response: Corrected, thank you for spotting it!
- Line 389: One “in” too much in the sentence.
Response: Corrected
- Line 569: in mice after “a” …
Response: Corrected
- Table 3: Ad-VEGF report first on Phase I and then on Phase II trial
Response: Corrected in Table 3, also in the text and references
- Maybe move “Immunodeficiency” paragraph behind the “Hematological diseases” as they both affect the hematopoietic system.
Responses: I understand the request. However, to place the “immunodeficiency” section after the section on “Hematological diseases” would require a major reconstruction of the whole manuscript.
- Line 636: Delete “)” behind miRNAs
Response: Corrected
- Table 5: Replace “1” with “I”
Response: Corrected
- Line 794: Delete “)”
Response: Corrected
- Line 810: another typo?
Response: Corrected
- Line 821: Delete “the” before highly attractive
Response: Corrected
Round 2
Reviewer 1 Report
The modifications introduced by the author have improved the accuracy of the manuscript. Still, some issues remain.
1. Previous point 2. I understand that the abstract has size constraints. However, citing only some examples and not others which are equally (or more) relevant causes a bias in this important section. This may be misleading for the reader. If all approved gene therapy products cannot be mentioned, I suggest the author cites the fields in which they are being applied (i.e., cancer, vaccination, hematological, metabolic and neurological diseases).
2. Previous point 3. The author supports the definition of gene therapy as “gene replacement” in a self-citation. However, replacement means that the mutated gene is eliminated and the correct copy is delivered in substitution. I believe this mechanism of action does not reflect the majority of gene therapy approaches. Among the thousands of gene therapy reports, how many of them really describe gene replacement? Gene supplementation, correction or modulation are the main mechanisms of action of current gene therapy.
3. Line 94. The persistence of gene expression mediated by High-Capacity Adenoviral vectors is not limited to one year. The longest published follow-up in non-human primates is 7 years (Brunetti-Pierri et al., HGT 2013).
4. Previous point 10. I do not question the expertise of the author in alphaviruses. However, the reference supporting the 8 Kb capacity of these vectors (Strauss and Strauss, Microbiol Rev 1994) actually indicates a cloning capacity of 4 Kb. The author should include a reference demonstrating the successful incorporation of 8 Kb inserts into standard alphavirus vectors.
5. Table 5 should include approved Covid-19 vaccines, not only Phase III trials.
Author Response
Reviewer 1, Round 2
The modifications introduced by the author have improved the accuracy of the manuscript. Still, some issues remain.
- Previous point 2. I understand that the abstract has size constraints. However, citing only some examples and not others which are equally (or more) relevant causes a bias in this important section. This may be misleading for the reader. If all approved gene therapy products cannot be mentioned, I suggest the author cites the fields in which they are being applied (i.e., cancer, vaccination, hematological, metabolic and neurological diseases).
Response: As suggested, the different indications have now been mentioned.
- Previous point 3. The author supports the definition of gene therapy as “gene replacement”in a self-citation. However, replacement means that the mutated gene is eliminated and the correct copy is delivered in substitution. I believe this mechanism of action does not reflect the majority of gene therapy approaches. Among the thousands of gene therapy reports, how many of them really describe gene replacement? Gene supplementation, correction or modulation are the main mechanisms of action of current gene therapy.
Response: Although the authors of Ref 266 (Russel et al. Lancet 2017, 390, 849-860) make the following statement in their abstract “Phase 1 studies have shown potential benefit of gene replacement in RPE65-mediated inherited retinal dystrophy” I am happy to follow the suggestion of replacing (haha!) “gene replacement” with “gene supplementation, correction or modulation”. Also, in the title of reference 234 “gene replacement” is used.
- Line 94. The persistence of gene expression mediated by High-Capacity Adenoviral vectors is not limited to one year. The longest published follow-up in non-human primates is 7 years (Brunetti-Pierri et al., HGT 2013).
Response: For clarification, there is not statement that adenovirus vector-based expression is limited to one year. Instead, it reads “at least one year”. However, the follow-up study by Brunetti-Pierri et al has been added and cited in references.
- Previous point 10. I do not question the expertise of the author in alphaviruses. However, the reference supporting the 8 Kb capacity of these vectors (Strauss and Strauss, Microbiol Rev 1994) actually indicates a cloning capacity of 4 Kb. The author should include a reference demonstrating the successful incorporation of 8 Kb inserts into standard alphavirus vectors.
Response: Reference 53 (Ehrengruber et al. Curr. Prot. Neurosci. 2011. 4.22.1-4.22.27) has been added.
- Table 5 should include approved Covid-19 vaccines, not only Phase III trials.
Response: The approvals (EUA) have been added to Table 5.